# STRETCHYSNAKE: FLEXIBLE VIDEOMAMBA FOR SHORT AND LONG-FORM ACTION RECOGNITION

## ABSTRACT

State space models (SSMs) have very recently been introduced as an alternative deep architecture to transformers, exhibiting competitive or superior performance across various language and vision tasks. However, both SSMs and transformers share certain limitations in the vision domain, namely spatio-temporal inflexibility. Traditionally, deep video models are trained on a fixed resolution and number of frames, often arbitrarily chosen as a trade-off between performance and computational cost. Changing the resolution and/or number of frames a model can ingest usually requires retraining the model, while avoiding re-training by variably changing the weights of a trained model leads to significantly reduced test accuracy. In this paper, we introduce a spatio-temporal flexible training method that encourages a single set of learned weights to adapt well to any input resolution or video length. We achieve this by simply randomly changing the spatial and temporal resolutions of a video during training, and dynamically interpolating the model's weights accordingly. This single change in training not only allows for one model to be applied to both short and long video understanding tasks alike, but also allows for user-specific tailoring of computational cost. We propose and evaluate 5 different spatio-temporal flexible training methods to find the optimal type for training a video SSM. We then evaluate our best flexibly-trained SSM, which we call StretchySnake, across a variety of short- and long-form action recognition evaluation protocols, such as video retrieval, fine-tuning, and linear probing, and massively outperform the same vanilla video SSM trained in a standard fashion by up to $28\%$ in some cases. Therefore, our training method can be used as a simple drop-in training technique for any SSM-based video models to strongly improve performance and instill spatio-temporal and compute flexibility.

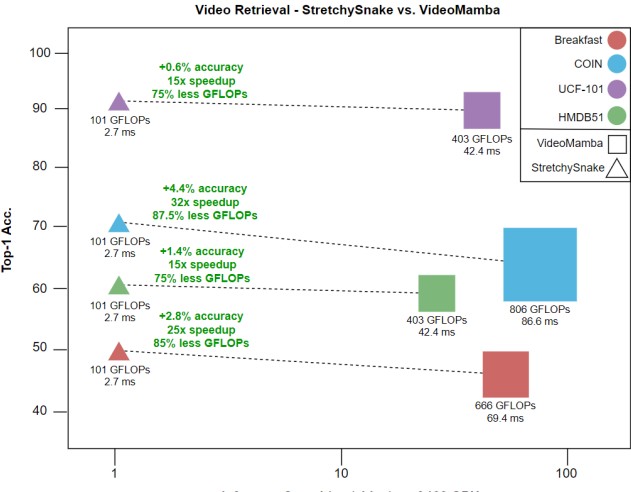

Figure 1: Comparing computational complexity and inference time between vanilla VideoMamba and StretchySnake. With StretchySnake's ability to accurately adapt to any spatial and temporal resolution, it can perform video retrieval at $8$ frames and $96$ pixels while *still significantly* outperforming the best VideoMamba in both accuracy and inference time and cost.

# 1 INTRODUCTION

The core goal of general video understanding is to learn high-quality spatio-temporal features that are robust to information redundancy in short-length videos and complex long-range dependencies in long videos alike. However, these two goals are often difficult to achieve unanimously with a single model due to the current practice of training video models. Video models are traditionally trained in a static fashion, wherein videos of a fixed length (referred to as temporal resolution) and frame size (referred to as spatial resolution) are fed as input. While this has sufficed as a design choice to balance computational complexity against performance, it severely limits the scalability and generalizability of video models. For example, previous state of the art (SOTA) image models trained in this static manner have been shown to suffer massive performance drops when tested at spatial resolutions unseen during training (Tian et al., 2023; Beyer et al., 2023). We show later that this phenomenon extends to video models as well, where current methods of training still perpetuate inflexibility in both the learned spatial and temporal features. Thus, in this paper we aim to tackle this issue by training a video model to learn a single set of weights that flexibly performs well on many spatial and temporal resolutions while mitigating degradations in test-time accuracy.

Currently, the transformer architecture (Vaswani, 2017; Dosovitskiy, 2020; Liu et al., 2022) has been dominant in every video domain, such as action recognition (Siddiqui et al., 2024), object segmentation (Kirillov et al., 2023), and large visual-language models (Zhu et al.; Lin et al., 2023). Despite transformers long-reigning supremacy in the field of computer vision, they are not without their limitations, mainly their quadratic-time complexity during training and inference. These issues have served as a significant barrier for transformers to learn extremely long-range dependencies, and thus limit their feasible application to important visual tasks such as long-form video understanding. To this extent, SSMs (Orvieto et al., 2023; Smith et al., 2022; Gu et al., 2021b) have very recently been proposed as an alternative architecture to transformers. Similar to the progression of transformers, SSMs were first observed to perform on-par, and in some cases outperform, transformer-based architectures in various natural language tasks (Gu et al., 2021a; 2020b) and were subsequently extended to images and videos (Chen et al., 2024; Li et al., 2024; Zhu et al., 2024). Most importantly to note, SSMs are much more adept at efficiently modeling long-range sequences due to their linear complexity, which is practically attainable during both training and testing (Gu et al., 2021a), and their ability to compress salient information across long contexts (Gu & Dao, 2023). However, we argue that adapting SSMs for video understanding in the same static fashion as video transformers severely underutilizes SSMs' long-range, context-dynamic capabilities (see Sec. 3.1).

In this paper, we propose a novel training method which dynamically changes the spatial and temporal resolutions of a video to equip a video SSM (VideoMamba (Li et al., 2024)) to learn better spatio-temporal features. However, certain layers and weights in a video model expect a fixed size input - such as a fixed frame size, number of frames, or embedding patch size - requiring on-the-fly adaptation to our variably sized input; a process which we call *spatio-temporal flexibility* (or st-flexibility). To achieve this, we interpolate the weights of said layers using differentiable transforms during training, enabling a video model to implicitly learn representations that are effective at various spatial and temporal scales. Additionally, st-flexibility can be implemented in a model through a variety of ways when dynamically changing spatial resolutions and lengths of videos during training (Sec. 4.2). Therefore, we introduce and evaluate 5 different versions of st-flexibility to ascertain the most effective type and train VideoMamba with the best method: our model we call StretchySnake.

Our main contributions are as follows:

- Introduce spatio-temporal flexible training, enabling a model to learn a single set of weights that performs well on all spatio-temporal resolutions *without* any architectural changes.

- Analysis of 5 different st-flexible methods across 4 action recognition datasets, gaining valuable insights on whether spatial or temporal resolution is more important for certain datasets.

- We find and train VideoMamba with the optimal version of st-flexibility, which massively outperforms vanilla VideoMamba on 4 action recognition datasets across various evaluation protocols.

- The computational efficiency of StretchySnake can be maximized by choosing the optimal balance between test accuracy and input resolution/length of video at test-time (Fig. 1).

## 2 RELATED WORKS

**Training Deep Learning Models Flexibly** Previous works have explored enabling an image model to generally perform well across multiple resolutions through a variety of means, like changing the model patch sizes/input resolutions (Beyer et al., 2023; Tian et al., 2023; Fan et al., 2024) or aspect ratios (Dehghani et al., 2024) during training. In a similar vein, other works have instilled multi-resolution capabilities in an image model by adopting a multi-stream approach (Xia et al., 2024; Yao et al., 2024; Tian et al., 2023), where training images are resized to different resolutions and simultaneously passed through separate branches to produce multi-scale features. However, this requires architectural changes and cannot be used as a drop-in training method for any model. Other works have extended similar ideas to videos, such as using multiple streams for different temporal resolutions (Zhang et al., 2023), using high temporal resolutions to efficiently "choose" only the important frames in a video (Zhang et al., 2022), or some combination of the two (Feichtenhofer et al., 2019). Tangential works have also shown that finding the optimal balance between input/model size and test accuracy is an exceptional way to optimize compute power (Alabdulmohsin et al., 2024).

Since our st-flexible method of training enables a model to perform well across a wide range of spatial and temporal resolutions, the optimal configuration for a fully-trained model can be chosen to minimize compute power without sacrificing significant performance (Fig. 1). Furthermore, it is important to note that our work separates itself in several ways: (1) we do not add any additional branches to the model to ingest variable spatial or temporal resolutions, but adaptively change the model on-the-fly during training, (2) we change the spatial and temporal resolutions of the input to learn better features and further show that our method of training enables the model to generally perform well across *all* spatio-temporal resolutions, and (3) we are the first to explore st-flexibility for video SSMs, since the aforementioned works only investigate attention-based models.

## 3 BACKGROUND

**State Space Models** Structured state space models (Gu et al., 2021a;b; 2022a) have shown great promise as efficient and powerful sequencing models. Broadly, their main attraction is their ability to be parameterized as either a convolution or recurrence, enabling GPU compatibility and near-linear scaling complexity with regards to sequence length. Traditionally, SSMs map some time-dependent, continuous input sequence of length $L$ into a latent state representation to predict the evolution of the latent state. Specifically, some input sequence $x(t) \in \mathbb{R}^L$ is mapped to some output sequence $y(t) \in \mathbb{R}^L$ through a learned latent state $h(t) \in \mathbb{R}^N$ of dimensionality $N$. SSMs learn this mapping through a two-stage sequence-to-sequence ordinary differential equation (ODE) consisting of four parameters $(\Delta, \mathbf{A}, \mathbf{B}, \mathbf{C})$:

$$h'(t) = \mathbf{A}h(t) + \mathbf{B}x(t) \tag{1}$$
$$y(t) = \mathbf{C}h(t) \tag{2}$$

where $\mathbf{A} \in \mathbb{R}^{N \times N}$ is the hidden state transition matrix and $\mathbf{B} \in \mathbb{R}^{1 \times N}$ and $\mathbf{C} \in \mathbb{R}^{N \times 1}$ are the input and output projection matrices, respectively. With this being a continuous process, a learnable step size $\Delta$ is introduced to discretize $\mathbf{A}$ and $\mathbf{B}$ with a variety of possibilities (Nguyen et al., 2022; Gu et al., 2022b), but we follow the zero-order hold used in (Gu & Dao, 2023):

$$\bar{\mathbf{A}} = \exp(\Delta\mathbf{A})$$
$$\bar{\mathbf{B}} = (\Delta\mathbf{A})^{-1}(\exp(\Delta\mathbf{A}) - \mathbf{I}) \cdot \Delta\mathbf{B}$$

After discretization, an SSM can be computed either as a linear recurrence (shown on the left) or a global convolution (as shown on the right):

$$h_t = \bar{\mathbf{A}}h_{t-1} + \bar{\mathbf{B}}x_t \qquad\qquad \bar{\mathbf{K}} = (\bar{\mathbf{C}}\bar{\mathbf{B}}, \bar{\mathbf{C}}\bar{\mathbf{A}}\bar{\mathbf{B}}, \bar{\mathbf{C}}\bar{\mathbf{A}}^2\bar{\mathbf{B}}, \cdots, \bar{\mathbf{C}}\bar{\mathbf{A}}^t\bar{\mathbf{B}})$$
$$y_t = \bar{\mathbf{C}}h_t \qquad\qquad\qquad y = x * \bar{\mathbf{K}}$$

Often times, the convolutional parameterization is chosen during training for parallelization, whereas the recurrent parameterization is used during inference for constant-time autoregression. There are other important SSM design choices that are currently being explored and optimized, such as initialization and structure of $\mathbf{A}$ (Gu et al., 2022a; Gupta et al., 2022; Smith et al., 2022) and linear time invariance (Peng et al., 2023; Fu et al., 2022) (or lack thereof (Gu & Dao, 2023)), but are not fully integral to understanding our work.

### 3.1 MOTIVATION FOR ST-FLEXIBILITY IN SSMs

Note that the matrix $\mathbf{A}$ (Eq. 1) in SSMs is of particular importance as it is responsible for the state-to-state transitions of the latent space - in other words, it compresses the cumulative history of all previously seen inputs at some timestep into a smaller latent state. It can be difficult to strike a balance between retaining salient information from older context in the model's memory, while still incorporating information from new context - especially so in extremely long contexts. To solve this issue, (Gu et al., 2020a) found that rather than initializing $\mathbf{A}$ randomly, it was crucial to initialize $\mathbf{A}$ following the HiPPO algorithm (Gu et al., 2020a) to enable SSMs to efficiently compress all previously seen history by simply learning the coefficients of a Legendre polynomial (Voelker et al., 2019). However, despite the near-linear complexity and compatibility with long-range dependencies, SSMs were still outclassed by attention-based mechanisms in one facet: the ability to focus or ignore particular inputs. Since attention does not compress data and instead ensures every token is attended to every other token, this quadratically-growing complexity is why transformers struggle to perform on extremely long contexts. Thus, (Gu & Dao, 2023) introduced a critical improvement to SSMs to enable them to perform content-aware reasoning across long contexts: the selective scan. By simply changing $\mathbf{B}$ and $\mathbf{C}$ to be functions of the input rather than being input-invariant, they can selectively keep or forget information as it propagates through the model. We hypothesize that this selective retention or forgetting of information (also known as "memory") is a major reason why st-flexibility massively improves performance in video SSM, as seen in Fig. 3 and further discussed in Sec. 5 and also the appendix. With regards to video understanding, constantly flexing the spatial and temporal resolutions of the video during training encourages the model to learn only the salient information at a variety of scales. Since $\bar{\mathbf{A}}, \bar{\mathbf{B}}, \bar{\mathbf{C}}$ are input dependent in VideoMamba, we hypothesize that training VideoMamba with inputs at a variety of spatio-temporal scales significantly improves the memorization of salient information, as opposed to the standard method of training at a fixed, singular spatio-temporal scale (discussed in Sec. 5.3).

## 4 METHODOLOGY

### 4.1 PRELIMINARIES

Consider some video:

$$\mathbf{x} \in \mathbb{R}^{T \times H \times W \times C} \tag{3}$$

where $(T, H, W, C)$ are the number of frames, height, width, and number of channels respectively. Typically, video models reduce each frame in a video into a sequence of $N = \sqrt{\frac{H \times W}{P \times P}}$ patches: $\mathbf{x}_n \in \mathbb{R}^{(P^2 \times C)}$, where $P$ is a pre-determined patch size such that $0 \equiv P \mod (H * W)$ and $n \in \{1, \dots N\}$. This process is referred to as *patchification* and is one way to control the amount of compute for video models. After patchification, the spatial embedding $\mathbf{E}_n^s$ is computed for each patch $\mathbf{x}_n$:

$$\mathbf{E}_n^s = conv(\mathbf{x}_n), \ \mathbf{E}_n^s \in \mathbb{R}^{\frac{H}{P} \times \frac{W}{P} \times D} \tag{4}$$

where $D$ is the chosen embedding size and $conv(\cdot)$ is either a 2-D or 3-D convolution operation. To account for permutation invariance in transformers and SSMs, a learned spatial positional embedding $\mathbf{E}_{pos} \in \mathbb{R}^{N \times D}$ is added to each patch embedding (after concatenation) to obtain the final spatial token representation for a single frame $\mathbf{z}^s$:

$$\mathbf{z}^s = (concat(\{\mathbf{E}_n^s, \ \forall n\}) + \mathbf{E}_{pos}) \in \mathbb{R}^{1 \times N \times D} \tag{5}$$

This per-frame process must also be applied to the temporal domain in order to be extended to videos. Subsequently, a learnable temporal positional embedding $\mathbf{E}_{tpos} \in \mathbb{R}^{1 \times N \times D}$ is added to every spatial token $\mathbf{z}^s$ corresponding to a single frame. Thus, the final temporal token representation $\mathbf{z}^t \in \mathbb{R}^{1 \times N \times D}$ for each frame in a video is obtained:

$$\mathbf{z}_j^t = \mathbf{z}_j^s + \mathbf{E}_{tpos} \tag{6}$$

for all $j \in \{1, \cdots, T\}$. Finally, a classification token $[CLS] \in \mathbb{R}^{1 \times D}$ meant to aggregate the learned information from all patch tokens is appended and used for downstream prediction (Devlin, 2018; Dosovitskiy, 2020). With the exception of some minor design choices (such as different types of spatio-temporal factorization), virtually every video-based model encodes videos in this manner before learning spatio-temporal representations (Fig. 2).

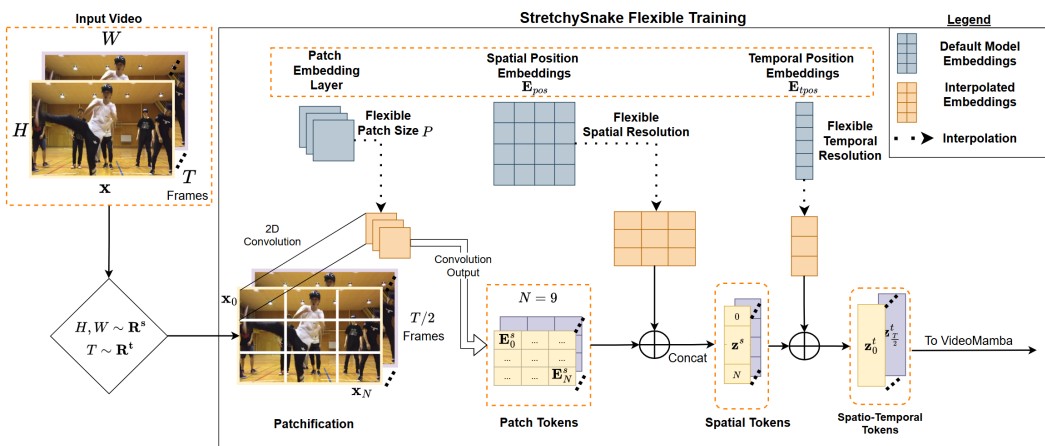

Figure 2: Our proposed method of training a video model with spatio-temporal flexibility. We highlight which tokens of a video model can be 'flexed' with dashed borders during training to accommodate for variable spatial and temporal resolutions in a video.

## 4.2 Instilling ST-Flexibility

The main goal of this work is to instill VideoMamba (and video SSMs in general) with *spatio-temporal flexibility*, or in other words, to learn a single set of weights that is robust to different spatial and temporal resolutions in a video. Ideally, a VideoMamba trained in this fashion would generally perform well during test-time across all types of videos (low vs. high resolution, short vs. long length, etc.) with minimal drops in performance.

Currently, the difficulty in training such a model is two-fold: (1) during training, certain layers and weights in the model must be interpolated accordingly to account for the changes in frame size and video length; and (2) the optimal method of instilling a model with st-flexibility is largely unexplored. Specifically, the convolutional embedding patch size (Eq. 4), number of spatial tokens (Eq. 5), and number of temporal tokens (Eq. 6) are the three key factors that dictate a model's capability to process videos of varying spatial and temporal lengths (Eq. 3). During training, these four equations can be changed (or *flexed*, as we refer to it from here on out) in many different combinations to allow for st-flexibility. In this work, we test 5 different versions of st-flexibility that can be applied to video models during training, which we list below. For all examples, assume the default model expects $T = 16$, $H = W = 224$ as input and $P = 16$ such that $N = \sqrt{\frac{224 \times 224}{16 \times 16}} = 14$, $\mathbf{E}_{pos} \in \mathbb{R}^{14 \times D}$, and $\mathbf{E}_{tpos} \in \mathbb{R}^{16 \times 14 \times D}$. For st-flexibility, spatial resolutions are sampled from the set $\mathbf{R^s} = \{96, 128, 224, 384\}$ and temporal resolutions are sampled from the set $\mathbf{R^t} = \{8, 16, 32, 64\}$.

1. **Temporal Flexibility**: Randomly sample $T$ during training from $\mathbf{R^t}$. Only flex the temporal tokens based on the number of input frames.

    **Example**: If $T \sim \mathcal{U}(\mathbf{R^t})$, assume for this example $T = 32$. Then, $\mathbf{x} \in \mathbb{R}^{32 \times 3 \times 224 \times 224}$, such that $\mathbf{E}_{tpos} \in \mathbb{R}^{16 \times N \times D}$ must be "flexed" to $\mathbf{E}_{tpos} \in \mathbb{R}^{32 \times N \times D}$

2. **Static Patch** - Randomly sample $T$ and $(H, W)$ during training from $\mathbf{R^t}$ and $\mathbf{R^s}$, respectively. Along with temporal flexibility, image size and number of spatial tokens are flexed, while the patch size is always kept static.

    **Example**: If $(H, W) \sim \mathcal{U}(\mathbf{R^s})$ and $T \sim \mathcal{U}(\mathbf{R^t})$, assume for this example that $T = 32$ and $H = W = 128$. Then, $\mathbf{x} \in \mathbb{R}^{32 \times 3 \times 128 \times 128}$ and fix $P = 16$ such that $N = \sqrt{\frac{128 \times 128}{16 \times 16}} = 8$ and $\mathbf{E}_{pos} \in \mathbb{R}^{16 \times D}$ must be "flexed" to $\mathbf{E}_{pos} \in \mathbb{R}^{8 \times D}$.

3. **Static Tokens**: Randomly sample $T$ and $(H, W)$ during training from $\mathbf{R^t}$ and $\mathbf{R^s}$, respectively. Along with temporal flexibility, image size and patch size are jointly flexed such that the resulting number of spatial tokens for every frame is always the same.

**Example**: If $(H, W) \sim \mathcal{U}(\mathbf{R^s})$ and $T \sim \mathcal{U}(\mathbf{R^t})$, assume for this example that $T = 32$ and $H = W = 128$. If $\mathbf{x} \in \mathbb{R}^{32 \times 3 \times 128 \times 128}$, then $P = 16$ must be "flexed" to $P = 9$ such that $N = \left\lfloor \sqrt{\frac{128 \times 128}{9 \times 9}} \right\rfloor = 14$ and $\mathbf{E}_{pos} \in \mathbb{R}^{14 \times D}$ does not need to be "flexed".

4. **FlexiViT**: Introduced in (Beyer et al., 2023) for images, fix $H = W = 240$ and randomly "flex" the patch size and number of spatial tokens from the pre-defined set in the original paper during training. Apply temporal flexing as described in the first example.

   **Example**: If $\mathbf{x} \in \mathbb{R}^{32 \times 3 \times 240 \times 240}$ and $P \sim \mathcal{U}(\{8, 10, 12, 15, 16, 20, 24, 30, 40, 48\})$, assume for this example that $P = 12$ such that $N = \sqrt{\frac{240 \times 240}{12 \times 12}} = 20$ and $\mathbf{E}_{pos} \in \mathbb{R}^{14 \times D}$ must be "flexed" to $\mathbf{E}_{pos} \in \mathbb{R}^{12 \times D}$.

5. **Flex-all**: Randomly sample $T$ and $(H, W)$ during training from $\mathbf{R^t}$ and $\mathbf{R^s}$, respectively. In addition to image size, convolution kernel size and number of spatial tokens are all flexed during training.

   **Example**: If $(H, W) \sim \mathcal{U}(\mathbf{R^s})$ and $T \sim \mathcal{U}(\mathbf{R^t})$, assume for this example that $T = 32$ and $H = W = 128$. Then, $\mathbf{x} \in \mathbb{R}^{32 \times 3 \times 128 \times 128}$, and choose $P$ such that $(0 \equiv P \bmod 128)$ and $12 \leq P \leq 48$. Assume for this example that $P = 32$ such that $N = \sqrt{\frac{128 \times 128}{32 \times 32}} = 4$ and $\mathbf{E}_{pos} \in \mathbb{R}^{14 \times D}$ must be "flexed" to $\mathbf{E}_{pos} \in \mathbb{R}^{4 \times D}$.

We postulate that training VideoMamba with this type of flexibility not only enables it to generalize to any size or length of video, but also results in better overall learned representations (Sec. 3.1). To flex the spatial resolution $(H, W)$ of a video we use the Resize function in PyTorch, and to flex the temporal resolution of a video $(T)$, we simply change the number of frames we uniformly sample in a training clip (Eq. 3). To flex the patch size of a model, we simply resize the weights $w$ of the patch embedding layer ($conv$ in Eq. 4) and the spatial positional embedding $\mathbf{E}_{pos}$ (Eq. 5) to the correct size using a 2-D bi-cubic interpolation. Lastly, we use a simple 1-D linear interpolation to flex the learned temporal positional embedding $\mathbf{E}_{tpos}$ to the correct size. Since all interpolation operations applied to $w$, $\mathbf{E}_{pos}$, and $\mathbf{E}_{tpos}$ are differentiable, their weights are still updated through backpropagation during st-flexible training.

## 5 EXPERIMENTS AND ABLATIONS

To validate that st-flexible training leads to better learned representations, we divide this section into 3 categories: (1) finding the optimal type of st-flexibility, (2) exhibiting the massive performance gains with StretchySnake over vanilla VideoMamba, and (3) comparing StretchySnake against SOTA action recognition baselines. To this extent, we examine these points using 3 types of transfer-learning experiments. Firstly, we perform video retrieval experiments on 4 benchmark action recognition datasets in total: 2 short-video action recognition datasets (UCF101 (Soomro, 2012) and HMDB51 (Kuehne et al., 2011)) and 2 long-video action recognition datasets (COIN (Tang et al., 2019) and Breakfast (Kuehne et al., 2014)) to evaluate our model with different context lengths (Table 1). Secondly, we perform fine-tuning and linear probing experiments on the same action recognition datasets (Table 2). Finally, we compare StretchySnake with previous SOTA uni-modal video models pre-trained solely on Kinetics-400 and show that StretchySnake outperforms every other model on average across all datasets in a video retrieval setting (Table 3). Moreover, StretchySnake can even outperform or competitively perform against multi-modal models which leverage additional modalities besides RGB or are pre-trained on additional data.

### 5.1 IMPLEMENTATION DETAILS

All experiments are performed by first training a VideoMamba on Kinetics-400 (Kay et al., 2017) exactly the same as a vanilla VideoMamba, but with st-flexibility. Specifically, we train with simple cross-entropy loss using the AdamW optimizer with 5 linear warm-up epochs. We use the default learning rate and weight decay values of $1e^{-3}$ and $0.05$, respectively. We initialize StretchySnake with the provided self-supervised pre-trained weights on Kinetics-400 (similarly done in (Tian et al., 2023)), and implement st-flexibility when performing further supervised training on Kinetics-400. We flexibly train for 12 epochs on Kinetics-400 and compare against a vanilla VideoMamba trained for 50 epochs, both in a supervised manner. In the fine-tuning experiments we further train the model pre-trained on Kinetics-400 on some downstream dataset, whereas in the linear probing experiments we freeze the entire pre-trained model and only train a linear classifier from scratch on the

downstream dataset. All of our experiments use VideoMamba-M, the largest sized VideoMamba as proposed in the original paper (Li et al., 2024) where $D = 576$. For temporal flexibility, we arbitrarily chose $\mathbf{R^t} = \{8, 16, 32, 64\}$. For all types of st-flexibility where applicable, we arbitrarily chose $\mathbf{R^s} = \{96, 128, 224, 384\}$. For FlexiViT, we follow their method by fixing $H = W = 240$ and randomly sampling from a set of patch sizes $\{8, 10, 12, 15, 16, 20, 24, 30, 40, 48\}$ during training. The vanilla baseline model we use for all comparisons was trained at a fixed temporal resolution of $T = 16$ and a fixed spatial resolution of $H = W = 224$. However, since VideoMamba provides weights for different versions of their model trained on Kinetics-400 at various temporal scales (8, 16, 32, and 64), we also provide even more comparisons in the appendix by separately comparing StretchySnake at each temporal scale against the corresponding vanilla VideoMamba. For certain st-flexible methods that train with variable patch sizes, we perform inference with a fixed patch size of 16 for fair comparisons to vanilla VideoMamba, but we provide extensive ablations with different patch sizes in the appendix. All experiments in this paper are performed on a single NVIDIA A100 80GB GPU.

## 5.2 FINDING THE OPTIMAL SPATIO-TEMPORAL FLEXIBILITY

To find the optimal type of st-flexibility for VideoMamba, we start by pre-training a VideoMamba model on Kinetics-400 with each of our proposed st-flexible methods. With the exception of st-flexibility, we follow the same exact protocol as baseline VideoMamba for supervised training on Kinetics-400. We then perform video retrieval across 4 different action recognition datasets, across different spatial and temporal resolutions, to find the best type of spatio-temporal flexibility. Figure 3 shows that at every temporal resolution and virtually every spatial resolution, static tokens appears to be the best performing and most robust type of st-flex for VideoMamba. For spatial resolutions $< 192$px, static-tokens massively outperforms the next best type of st-flexibility, usually in some range between $1\% - 18\%$. For spatial resolutions $> 192$px, static tokens still either outperforms or is on-par with other st-flexible methods in almost every setting, and only underperforms compared to other st-flexible methods in very rare cases (only on the Breakfast dataset at low/medium spatial and temporal resolutions). Importantly to note, not only does every st-flexible method outperform vanilla VideoMamba, as expected, but they also outperform vanilla VideoMamba at its default configuration of $T = 16$ and $H = W = 224$. Thus, we conclude that the best type of st-flexibility from our proposed methods is static-tokens, and we refer to this best model as **StretchySnake**.

## 5.3 STRETCHYSNAKE BEATS VANILLA VIDEOMAMBA

### 5.3.1 QUANTITATIVE RESULTS

With static tokens established as the optimal type of st-flexible method, we perform the same video retrieval experiments as Sec. 5.2 with vanilla VideoMamba for comparison with StretchySnake. Table 1 exhibits how StretchySnake beats vanilla VideoMamba at **every** spatial and temporal resolution, both seen and unseen during training, including vanilla VideoMamba's original configuration ($T = 16$, $H = W = 224$). Consistent double-digit improvements are observed in nearly every setting, across every dataset, over vanilla VideoMamba. The largest improvements on the long-video datasets (COIN and Breakfast) occur at the higher temporal resolutions, due to their specific need for long-context understanding. With the highest average improvement across all datasets being on the 64-frame setting of Breakfast at $24.8\%$, st-flexibility seems to strongly improve the long-range understanding of VideoMamba. Conversely, the largest improvements with respect to the short-video datasets (UCF101 and HMDB51) are seen at the lower 8-frame and 16-frame temporal resolution scales. Important to note is the relative stability of StretchySnake across all spatial and temporal resolutions alike, as compared to the drastic drops in performance of vanilla VideoMamba across different spatial resolutions. Interestingly, vanilla VideoMamba seems to be relatively stable when only changing the number of frames during evaluation and keeping $H = W = 224$. However, StretchySnake appears to leverage the additional information when increasing temporal resolution much more effectively than VideoMamba, as seen in StretchySnake's consistent improvements with increasing temporal resolution on the long-video COIN and Breakfast datasets; a behavior not similarly observed with vanilla VideoMamba. Thus, StretchySnake (and by extension, any model trained with st-flexibility) is much better equipped to adapt to the optimal temporal and spatial resolution for specific datasets as opposed to standardly trained models.

In Table 2, we further compare vanilla VideoMamba and StretchySnake in the additional transfer learning settings of fine-tuning and linear probing. The linear probing results are another testament

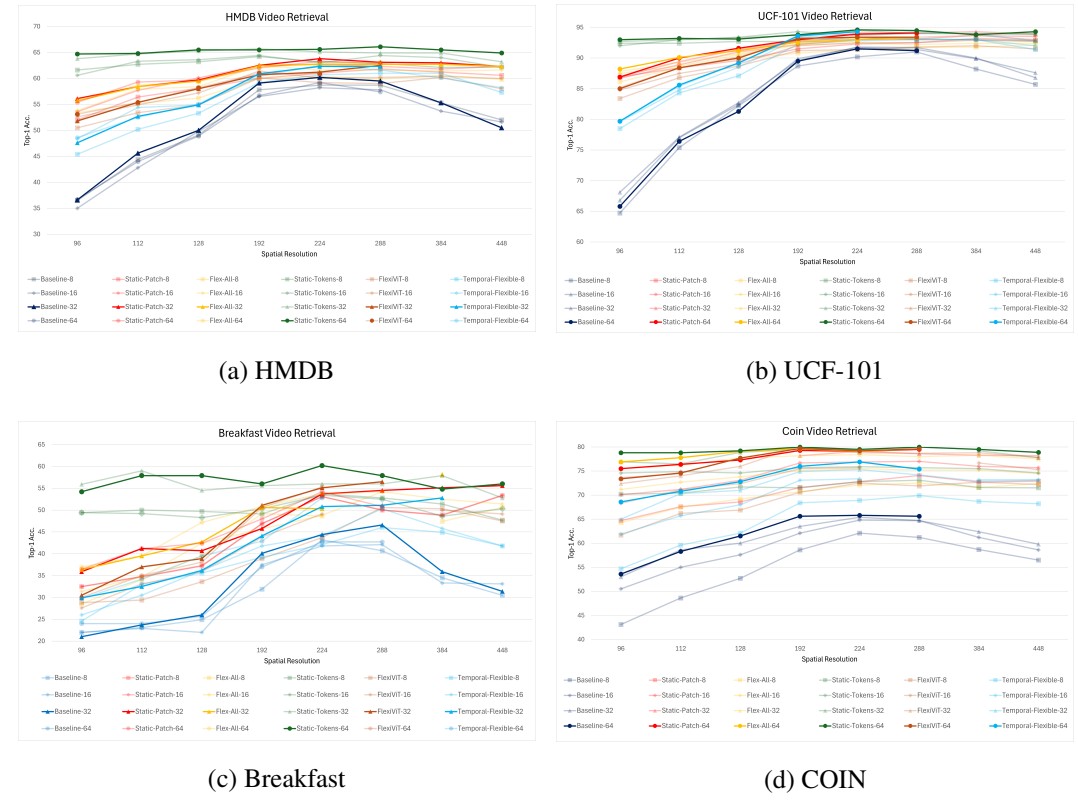

(a) HMDB

(b) UCF-101

(c) Breakfast

(d) COIN

Figure 3: Each graph best viewed with zoom. Video retrieval results on all four datasets at various spatial and temporal resolutions at test time. In every dataset, at virtually every configuration, static-tokens is the best performing method of spatio-temporal flexibility. The suffix ($-8$, $-16$, etc.) and marker for each label in the legend denotes temporal resolution. For better visibility, only the best-performing setting for each method is bolded.

to StretchySnake's superior learned representations, as freezing the model and simply only training a linear classifier still leads to significant improvements across every dataset, with only a marginal improvement on HMDB51. Fine-tuning is a less direct comparison of learned representations than linear probing, since in this setting both models are entirely unfrozen and trained using the standard, fixed method of training video models. Despite this, after training both models with $T = 16, H = W = 224$ for 30 epochs, StretchySnake's weights serve as a better quality initialization point in this setting as indicated by the uniform improvements across every dataset over vanilla VideoMamba.

### 5.3.2 QUALITATIVE RESULTS

We also qualitatively explore StretchySnake at both the feature and classification levels to visualize the improved representations of StretchySnake. In Fig. 4, we visualize the [CLS] token of both vanilla VideoMamba and StretchySnake on UCF101 at the lowest spatial scale and fix $T = 16$. StretchySnake still produces stable features at even the lowest spatial resolution on unseen data, leading to the consistently higher and stable video retrieval top-1 accuracy of StretchySnake seen in Table 1. In addition to better clustered [CLS] tokens, we also visualize the superior patch features from the penultimate layer of StretchySnake (Fig. 5). The patch features are the tokens from the last layer that are often discarded since the singular [CLS] token, which is meant to be an aggregation of all patch tokens, is used commonly used for predictions (Bertasius et al., 2021; Dosovitskiy, 2020). However, the final patch features contain more granular information to investigate the spatial activations of a video model at each frame (Oquab et al., 2023). Many additional visualizations can be found in the appendix.

### 5.4 FLEXIBLE VIDEOMAMBA BEATS SOTA MODELS

In addition to StretchySnake's improved video understanding capabilities over vanilla VideoMamba, we further compare against current SOTA methods in short- and long-video action recognition pre-

Table 1: Comparing vanilla VideoMamba performance with StretchySnake. Cells highlighted in gray are seen during training, with "VideoMamba$_{fx}$" denoting the number of frames used during evaluation. Best vanilla VideoMamba results are in red, with StretchySnake best results in green. StretchySnake outperforms baseline VideoMamba in virtually every setting, even at unseen resolutions and length of videos. Vanilla VideoMamba encounters out-of-memory (OOM) errors at large temporal and spatial resolutions due to its static patch size, while StretchySnake's adaptability prevents this issue.

| Dataset | Model | Testing Spatial Resolutions | | | | | | | | Avg. Δ% |
|---|---|---|---|---|---|---|---|---|---|---|
| | | 96 | 112 | 128 | 192 | 224 | 288 | 384 | 448 | |
| Breakfast | VideoMamba$_{f8}$ | 22.0 | 23.1 | 24.9 | 31.9 | 43.2 | 40.7 | 34.5 | 30.5 | - |
| | StretchySnake$_{f8}$ | 49.4 | 50.0 | 49.7 | 49.1 | 53.7 | 52.8 | 51.4 | 47.7 | +19.1 |
| | VideoMamba$_{f16}$ | 22.0 | 22.9 | 22.0 | 37.5 | 41.8 | 42.1 | 33.3 | 33.1 | - |
| | StretchySnake$_{f16}$ | 49.4 | 49.2 | 48.3 | 50.3 | 53.4 | 52.5 | 48.6 | 50.3 | +18.4 |
| | VideoMamba$_{f32}$ | 20.6 | 23.7 | 26.0 | 40.1 | 44.4 | 46.6 | 35.9 | 31.4 | - |
| | StretchySnake$_{f32}$ | 55.9 | 56.0 | 54.5 | 55.6 | 56.0 | 56.0 | 59.0 | 52.8 | +22.1 |
| | VideoMamba$_{f64}$ | 23.4 | 24.0 | 25.7 | 37.0 | 42.7 | 42.7 | OOM | OOM | - |
| | StretchySnake$_{f64}$ | 54.2 | 57.9 | 57.9 | 56.0 | 60.2 | 57.9 | 54.8 | 56.0 | +24.8 |
| COIN | VideoMamba$_{f8}$ | 43.1 | 49.5 | 52.7 | 58.6 | 62.1 | 61.2 | 58.7 | 56.5 | - |
| | StretchySnake$_{f8}$ | 70.2 | 70.4 | 71.7 | 71.5 | 72.8 | 73.1 | 71.6 | 71.5 | +16.3 |
| | VideoMamba$_{f16}$ | 50.5 | 55.0 | 57.6 | 62.1 | 64.8 | 64.7 | 61.2 | 58.6 | - |
| | StretchySnake$_{f16}$ | 74.6 | 74.9 | 74.6 | 75.7 | 75.9 | 75.7 | 75.5 | 74.6 | +13.6 |
| | VideoMamba$_{f32}$ | 53.0 | 58.6 | 60.0 | 63.5 | 65.4 | 64.7 | 62.4 | 59.8 | - |
| | StretchySnake$_{f32}$ | 76.9 | 76.5 | 78.9 | 79.5 | 79.0 | 79.4 | 79.2 | 77.8 | +17.5 |
| | VideoMamba$_{f64}$ | 53.6 | 58.3 | 61.5 | 65.6 | 65.8 | 65.6 | OOM | OOM | - |
| | StretchySnake$_{f64}$ | 78.8 | 78.8 | 79.2 | 80.0 | 79.5 | 80.0 | 79.5 | 78.9 | +17.7 |
| UCF-101 | VideoMamba$_{f8}$ | 64.7 | 75.4 | 82.2 | 88.7 | 90.2 | 91.0 | 88.2 | 85.7 | - |
| | StretchySnake$_{f8}$ | 92.4 | 92.4 | 92.7 | 92.7 | 93.4 | 93.1 | 93.0 | 92.8 | +16.8 |
| | VideoMamba$_{f16}$ | 66.8 | 77.0 | 82.4 | 89.9 | 91.7 | 91.4 | 89.9 | 87.6 | - |
| | StretchySnake$_{f16}$ | 92.0 | 93.0 | 93.4 | 94.3 | 93.4 | 94.0 | 94.0 | 93.8 | +8.9 |
| | VideoMamba$_{f32}$ | 68.1 | 77.1 | 82.7 | 89.6 | 91.8 | 91.7 | 90.0 | 86.8 | - |
| | StretchySnake$_{f32}$ | 92.7 | 93.0 | 93.3 | 93.4 | 93.9 | 94.0 | 94.0 | 94.0 | +8.8 |
| | VideoMamba$_{f64}$ | 65.8 | 76.4 | 81.3 | 89.5 | 91.5 | 91.2 | OOM | OOM | - |
| | StretchySnake$_{f64}$ | 93.0 | 93.2 | 93.1 | 93.6 | 94.3 | 94.5 | 93.8 | 94.3 | +11.0 |
| HMDB-51 | VideoMamba$_{f8}$ | 36.5 | 44.4 | 49.1 | 57.8 | 58.7 | 58.7 | 55.3 | 52.0 | - |
| | StretchySnake$_{f8}$ | 61.6 | 62.7 | 63.2 | 64.2 | 63.2 | 62.9 | 62.1 | 62.2 | +15.3 |
| | VideoMamba$_{f16}$ | 35 | 42.8 | 49.8 | 56.5 | 58.2 | 57.8 | 53.7 | 51.6 | - |
| | StretchySnake$_{f16}$ | 60.6 | 63.3 | 63.6 | 64.4 | 63.0 | 64.4 | 64.0 | 62.1 | +12.5 |
| | VideoMamba$_{f32}$ | 36.6 | 45.6 | 50.0 | 59.1 | 60.2 | 59.5 | 55.3 | 50.5 | - |
| | StretchySnake$_{f32}$ | 63.8 | 64.7 | 65.3 | 65.7 | 65.1 | 64.9 | 64.9 | 63.2 | +12.6 |
| | VideoMamba$_{f64}$ | 36.7 | 44 | 48.9 | 56.7 | 59.2 | 59.0 | OOM | OOM | - |
| | StretchySnake$_{f64}$ | 64.7 | 64.8 | 65.5 | 65.5 | 65.6 | 66.1 | 65.5 | 64.9 | +11.0 |

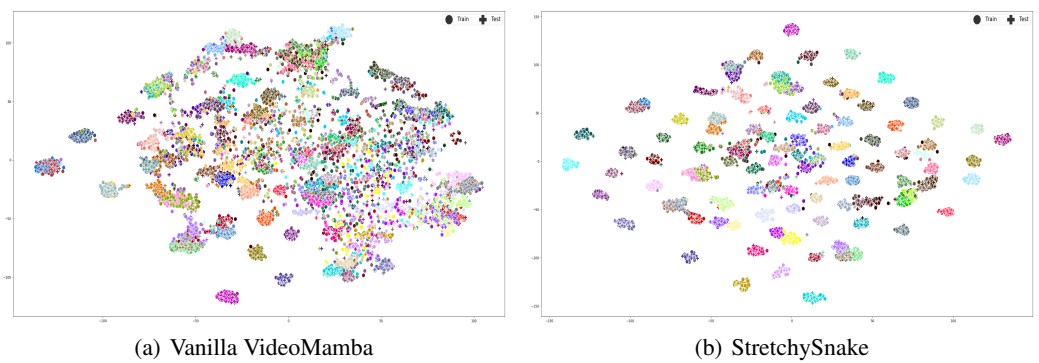

(a) Vanilla VideoMamba      (b) StretchySnake

Figure 4: Visualizing video retrieval using a t-SNE plot of the [CLS] token taken from the last layer during evaluation on the UCF-101 dataset with $H = W = 96$ pixels and $T = 16$ frames. StretchySnake accurately clusters action classes at low resolutions where the vanilla model fails, clearly exhibiting its robustness to changes in spatial resolutions even on unseen data. Additional visualizations at different spatial scales are provided in the appendix.

trained on Kinetics-400. Across four action recognition datasets, StretchySnake performs the best on average, and in some cases outperforms multi-modal models or models trained on extra data. Thus, training VideoMamba with st-flexibility greatly increases the quality of its learned representations, and moreover, better leverages VideoMamba's dynamic context length modeling capabilities.

Table 2: Comparing vanilla VideoMamba with StretchySnake across four action recognition datasets. We report results on full-finetuning (the entire model is trained on the respective dataset) and linear probing (the model is frozen and only a linear classifier is trained).

| Model | Full Finetuning | | | | Linear Probing | | | |
|---|---|---|---|---|---|---|---|---|
| | UCF101 | HMDB51 | COIN | Breakfast | UCF101 | HMDB51 | COIN | Breakfast |
| VideoMamba | 95.7 | 75.0 | 84.0 | 82.6 | 89.1 | 63.6 | 75.5 | 58.6 |
| Ours | 96.5 (+0.8) | 76.9 (+1.9) | 88.1 (+4.1) | 86.8 (+4.2) | 94.1 (+5.0) | 64.0 (+0.4) | 80.7 (+5.2) | 62.8 (+4.2) |

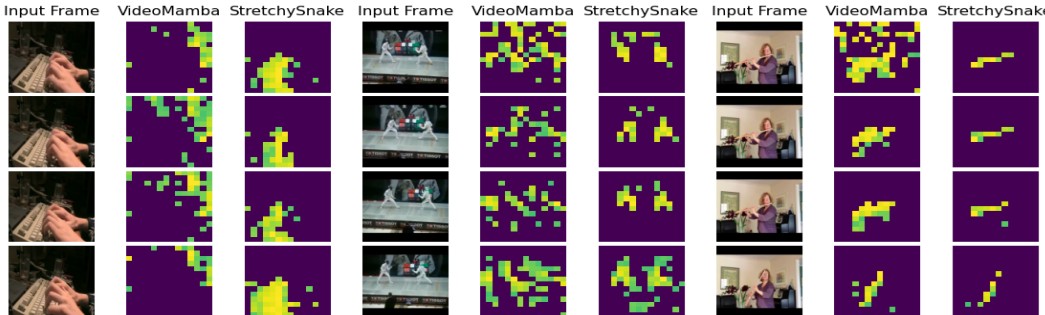

Figure 5: Visualizing frame activations between VideoMamba and StretchySnake on random UCF videos. For fair comparisons we set $T = 16$ and $H = W = 224$, and for brevity we randomly show 4 frames from the video. Not only does StretchySnake localize and activate on the correct region in the frame better than vanilla VideoMamba (left, middle), but it also does so in fewer frames (right).

Table 3: Comparing video retrieval results with previous SOTA methods. StretchySnake massively outperforms vanilla VideoMamba and also performs the best across both short- and long-form action recognition datasets compared to previous SOTA methods trained on Kinetics-400. Best **unimodal** results are in green, with second best in red. Gray results denotes the model was trained on additional modalities besides RGB (‡) or extra data (†).

| Model | # of Params | Video Retrieval | | | | |
|---|---|---|---|---|---|---|
| | | UCF101 (EP1) | HMDB51 | COIN | Breakfast | Average |
| Uniformer (Li et al., 2023b)(ICLR '22) | 49.8M | 87.4 | 53.4 | 44.1 | 22.9 | 52.0 |
| MViT (Fan et al., 2021)(ICCV '21) | 36.0M | 87.2 | 57.7 | 48.0 | 28.0 | 55.2 |
| Hiera-B (Ryali et al., 2023)(ICML '23) | 51.1M | 94.3 | 64.0 | 61.3 | 42.1 | 65.4 |
| VideoMamba (Li et al., 2024)(ECCV '24) | 73.8M | 91.8 | 60.2 | 65.8 | 46.3 | 66.0 |
| TimeSFormer (Bertasius et al., 2021)(ICML '21) | 121.5M | 91.6 | 58.7 | 76.3 | 39.5 | 66.5 |
| VideoSwin (Liu et al., 2022)(CVPR '22) | 88.0M | 93.9 | 58.9 | 65.8 | 52.3 | 67.7 |
| Hiera-L (Ryali et al., 2023)(ICML '23) | 213.1M | 96.4 | 66.0 | 64.5 | 50.2 | 69.4 |
| CAST (Lee et al., 2024)(NeurIPS '23) | 45.3M | 95.0 | 65.0 | 75.1 | 49.7 | 71.2 |
| EVL (Lin et al., 2022)(ECCV '22)‡ | 33.2M | 94.4 | 61.9 | 81.0 | 42.3 | 69.9 |
| Omnivore (Girdhar et al., 2022)(CVPR '22)† | 90.1M | 95.1 | 62.3 | 71.2 | 53.9 | 70.6 |
| UniformerV2 (Li et al., 2023a)(ICCV '23)‡ | 114.5M | 95.2 | 65.6 | 78.7 | 48.5 | 72.0 |
| AIM (Yang et al., 2023)(ICLR'23)‡ | 96.4M | 94.5 | 66.0 | 82.8 | 54.2 | 74.4 |
| Ours | 73.8M | 94.5 | 66.1 | 80.0 | 60.2 | 75.2 |

## 6 CONCLUSION

In this paper, we propose a novel method of training video models to instill spatio-temporal flexibility. During training, we dynamically change the frame size and length of a video to better enable a deep video model to perform well across a vast range of spatial and temporal resolutions. With the variety of combinations with which st-flexibility can be implemented in a model during training, we propose and analyze five different spatio-temporal methods to find the optimal type. Moreover, we apply our best method of training to the video-SSM model VideoMamba, calling this model StrechySnake, and show that st-flexibility massively improves downstream performance across multiple short- and long-form action recognition datasets. With performance gains as high as 28% over vanilla VideoMamba, we effectively demonstrate that StrechySnake contains better quality representations at all spatial and temporal scales; an especially valuable quality given SSM's propensity for learning better long-range dependencies. Additionally, our training method allows for the choice to use any spatial or temporal resolution at inference time without major degradation in performance, accommodating any computational budget.

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
