# A    APPENDIX

## A.1    OVERVIEW

We organize the appendix into the following sections:

- Sec. A.2 further describes the evaluation protocols used in this paper for clarity.
- Sec. A.3 ablates different patch sizes during evaluation for certain types of st-flexibility which are trained with many different patch sizes.
- Sec. A.4 provides even deeper video retrieval comparisons between StretchySnake and different versions of VideoMamba trained at specific temporal resolutions.
- Lastly, Sec. A.5 provides additional qualitative visualizations as shown in the main paper, namely more [CLS] token and patch activation map visualizations across different datasets and spatial resolutions.

## A.2    EVALUATION DETAILS

A majority of our experiments focus on feature evaluation, where the learned representations of our models are frozen and evaluated through a variety of downstream datasets and evaluation protocols. Following previous works such as (Han et al., 2020; Dave et al., 2024; Diba et al., 2021), performing video retrieval and linear probing experiments are effective and appropriate means to evaluate how our proposed st-flexible method leads to better learned representations. For clarity, video retrieval consists of taking all training samples from a dataset and extracting features for each sample (called a gallery) using a pre-trained, frozen model. Then, each test sample (called a query) is passed through the same model and the resulting feature is compared with every feature from the gallery. The nearest neighbor is 'retrieved', and the top-1 accuracy we report throughout the paper is how many test samples and their corresponding top-1 retrieval have the same action label, which is standard protocol. Fine-tuning and linear probing details can be found in the main paper.

## A.3    ABLATIONS WITH DIFFERENT PATCH SIZES FOR FLEX-ALL AND FLEXIVIT

An important ablation is examining the effect of patch size during evaluation for the "Flex-All" and "FlexiViT" methods of st-flexibility. Since these methods train with a dynamically changing patch size, we set the patch size $ps = 16$ for all experiments in the main paper for fair comparisons to vanilla VideoMamba. However, one could argue that static-tokens is only outperforming these methods due to its adaptive patch size at test time, especially at extremely low or high resolutions. Thus, we provide video retrieval results in Table A1 where we compare static-tokens, Flex-All, and FlexiViT at the same patch sizes. Specifically, we set Flex-All and FlexiViT to use whatever patch size static-tokens would use in the same scenario and show that static-tokens is still the best performing method of st-flexibility.

Table A1: Ablating different patch sizes during evaluation with Flex-All and FlexiViT. Even when using the same patch size as static-tokens, Flex-All and FlexiViT don't reach the same level of performance, further supporting that static-tokens is generally the best method of st-flexible training. Best results are in **bold**.

| Dataset | St-Flexible Method | Testing Spatial Resolutions/Patch Size | | | | | | | |
| --- | --- | --- | --- | --- | --- | --- | --- | --- | --- |
| | | 96/7 | 112/8 | 128/9 | 192/14 | 224/16 | 288/21 | 384/27 | 448/32 |
| Breakfast | FlexiViT | 42.6 | 46.0 | 46.3 | 48.3 | 49.1 | 46.3 | 48.3 | 44.9 |
| | Flex-All | 38.7 | 45.4 | 47.0 | 46.9 | 48.8 | 50.2 | 46.0 | 48.6 |
| | Static-Tokens | **49.4** (+6.8) | **49.2** (+3.2) | **48.3** (+1.3) | **50.3** (+2.0) | **53.4** (+4.3) | **52.5** (+2.3) | **48.6** (+0.3) | **50.3** (+1.7) |
| COIN | FlexiViT | 70.7 | 73.6 | 74.4 | 75.2 | 75.7 | 75.5 | 74.7 | 73.6 |
| | Flex-All | 69.4 | 72.3 | 73.8 | 74.9 | 75.5 | 75.3 | 75.4 | 73.9 |
| | Static-Tokens | **74.6** (+3.9) | **74.9** (+1.3) | **74.6** (+0.2) | **75.7** (+0.5) | **75.9** (+0.2) | **75.7** (+0.2) | **75.5** (+0.1) | **74.6** (+0.7) |
| UCF-101 | FlexiViT | 89.4 | 91.2 | 91.3 | 92.3 | 92.5 | 92.4 | 91.7 | 91.3 |
| | Flex-All | 89.0 | 90.5 | 91.4 | 92.3 | 92.4 | 92.4 | 92.0 | 91.8 |
| | Static-Tokens | **92.0** (+2.6) | **93.0** (+1.8) | **93.4** (+2.0) | **94.3** (+2.0) | **93.4** (+0.9) | **94.0** (+1.6) | **94.0** (+2.0) | **93.8** (+2.0) |
| HMDB-51 | FlexiViT | 55.3 | 57.1 | 59.9 | 61.7 | 60.1 | 61.8 | 60.7 | 59.1 |
| | Flex-All | 54.8 | 59.0 | 60.1 | 61.0 | 61.6 | 61.7 | 61.3 | 60.7 |
| | Static-Tokens | **60.6** (+5.3) | **63.3** (+4.3) | **63.6** (+3.5) | **64.4** (+2.7) | **63.0** (+1.4) | **64.4** (+2.6) | **64.0** (+2.7) | **62.1** (+1.6) |

### A.4 DEEPER VIDEO RETRIEVAL COMPARISONS BETWEEN STRETCHYSNAKE AND VIDEOMAMBA

In the main paper, we perform all experiments with a vanilla VideoMamba trained on Kinetics-400 at $T = 16$ and $H = W = 224$. Thus, Table 1 in the main paper evaluates VideoMamba's performance on unseen spatial AND temporal resolutions. Moreover, since StretchySnake is trained with variable temporal resolutions, it may have an inherit edge against vanilla VideoMamba when we change the temporal resolution in Table 1 of the main paper. Thus, we provide results in Table A2 where we compare StretchySnake with a vanilla VideoMamba trained at the same temporal resolution that is used during evaluation. We simply load the different weights of VideoMamba trained on Kinetics-400 at $T = 8$, $T = 32$, and $T = 64$ originally provided by the authors. For example, we perform video retrieval with $T = 8$ on the Breakfast dataset in rows $1 - 2$ and compare StretchySnake with a vanilla VideoMamba trained on Kinetics-400 at $T = 8$ and $H = W = 224$. Similarly, we perform video retrieval with $T = 32$ on the Breakfast dataset in rows $3 - 4$ and compare StretchySnake with a vanilla VideoMamba trained on Kinetics-400 at $T = 32$ and $H = W = 224$. Essentially, this is a fairer baseline since we are comparing against vanilla VideoMambas that are performing video retrieval at the same temporal resolution they were trained on. However, StretchySnake still heavily outperforms these models in every setting, further exemplifying StretchySnake's adaptability to any spatio-temporal resolution.

Table A2: Comparing StrechySnake and vanilla VideoMambas trained at the same temporal resolution used during evaluation. Cells highlighted in gray are seen during training. Best vanilla VideoMamba results are in red, with best StretchySnake results in green.

| Dataset | Model | Testing Spatial Resolutions | | | | |
|---|---|---|---|---|---|---|
| | | 112 | 192 | 224 | 288 | 448 |
| Breakfast | VideoMamba ($T = 8$) | 17.7 | 38.1 | 35.5 | 42.0 | 26.5 |
| | StretchySnake ($T = 8$) | **50.0** | **49.1** | **53.7** | **52.8** | **47.7** |
| | VideoMamba ($T = 32$) | 24.2 | 40.6 | 42.1 | 46.9 | 32.4 |
| | StretchySnake ($T = 32$) | **56.0** | **55.6** | **56.0** | **56.0** | **52.8** |
| | VideoMamba ($T = 64$) | 22.0 | 42.9 | 46.8 | 46.8 | 33.2 |
| | StretchySnake ($T = 64$) | **57.9** | **56.0** | **60.2** | **57.9** | **56.0** |
| COIN | VideoMamba ($T = 8$) | 50.1 | 60.5 | 65.2 | 63.5 | 58.5 |
| | StretchySnake ($T = 8$) | **70.4** | **71.5** | **72.8** | **73.1** | **71.5** |
| | VideoMamba ($T = 32$) | 58.5 | 65.8 | 67.9 | 66.2 | 61.6 |
| | StretchySnake ($T = 32$) | **76.5** | **79.5** | **79.0** | **79.4** | **77.8** |
| | VideoMamba ($T = 64$) | 59.9 | 66.1 | 66.4 | 67.6 | 64.3 |
| | StretchySnake ($T = 64$) | **78.8** | **80.0** | **79.5** | **80.0** | **78.9** |
| UCF-101 | VideoMamba ($T = 8$) | 76.1 | 88.3 | 90.1 | 90.7 | 86.3 |
| | StretchySnake ($T = 8$) | **92.4** | **92.7** | **93.4** | **93.1** | **92.8** |
| | VideoMamba ($T = 32$) | 79.5 | 90.0 | 92.5 | 92.4 | 87.7 |
| | StretchySnake ($T = 32$) | **93.0** | **93.4** | **93.9** | **94.0** | **94.0** |
| | VideoMamba ($T = 64$) | 80.1 | 90.7 | 92.5 | 92.5 | 89.9 |
| | StretchySnake ($T = 64$) | **93.2** | **93.6** | **94.3** | **94.5** | **94.3** |
| HMDB-51 | VideoMamba ($T = 8$) | 41.2 | 56.5 | 57.6 | 56.3 | 47.6 |
| | StretchySnake ($T = 8$) | **62.7** | **64.2** | **63.2** | **62.9** | **62.2** |
| | VideoMamba ($T = 32$) | 47.1 | 60.8 | 62.7 | 62.5 | 53.4 |
| | StretchySnake ($T = 32$) | **64.7** | **65.7** | **65.1** | **64.9** | **63.2** |
| | VideoMamba ($T = 64$) | 47.8 | 61.6 | 62.7 | 62.8 | 59.7 |
| | StretchySnake ($T = 64$) | **64.8** | **65.5** | **65.6** | **66.1** | **64.9** |

### A.5 ADDITIONAL QUALITATIVE VISUALIZATIONS

In this section, we provide additional visualizations as seen in the main paper at different spatial resolutions and on different datasets. For fairest comparisons to Vanilla VideoMamba, in all experiments we fix $T = 16$ and only visualize resolutions that are unseen during training to both StretchySnake and vanilla VideoMamba. We include $H = W = 224$ to show StretchySnake's improvement over

vanilla VideoMamba even at the default setting. It is important to note that StretchySnake and vanilla VideoMamba are only trained on the Kinetics-400 dataset, so all visualizations are on completely unseen data.

### A.5.1 [CLS] T-SNE

In this section, we show additional [CLS] token visualizations at different spatial resolutions during video retrieval, where each color denotes one class (with some redundancy due to the high number of classes in both datasets). We show results on one short action recognition dataset (UCF-101, Figs. A1-A4) and one long action recognition dataset (COIN, Figs. A5-A8). Each graph best viewed with zoom.

On the UCF-101 dataset (Figs. A1-A4), StretchySnake produces stable, consistent features across both low and high spatial resolution scales, whereas VideoMamba not only struggles to cluster each action at low spatial resolutions, but still does not achieve the same level of clustering as StrechySnake even at high spatial resolutions ($H = W = 448$).

On the COIN dataset (Figs. A5-A8), since both models do not achieve the same high levels of accuracy as UCF-101 ($> 90\%$), there is significantly more noise in the visualizations. Despite this, these visualizations serve to similarly show StrechySnake's more stable performance across a variety of unseen spatial resolutions when compared to vanilla VideoMamba. For example, vanilla VideoMamba has considerably more inter-class variation than StretchySnake, specifically at the unseen spatial resolutions ($H = W = \{112, 192, 448\}$).

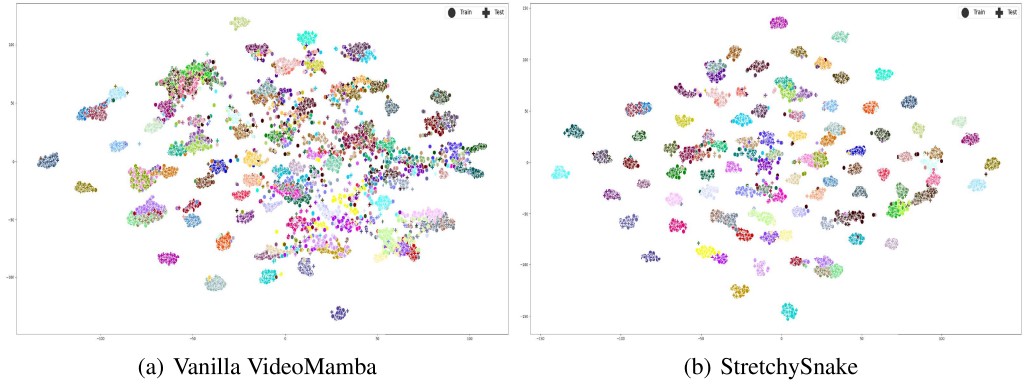

(a) Vanilla VideoMamba                                    (b) StretchySnake

Figure A1: [CLS] token visualization on the UCF-101 dataset at $H = W = 112$ pixels. Each color denotes one class (with some redundancy due to the high number of classes).

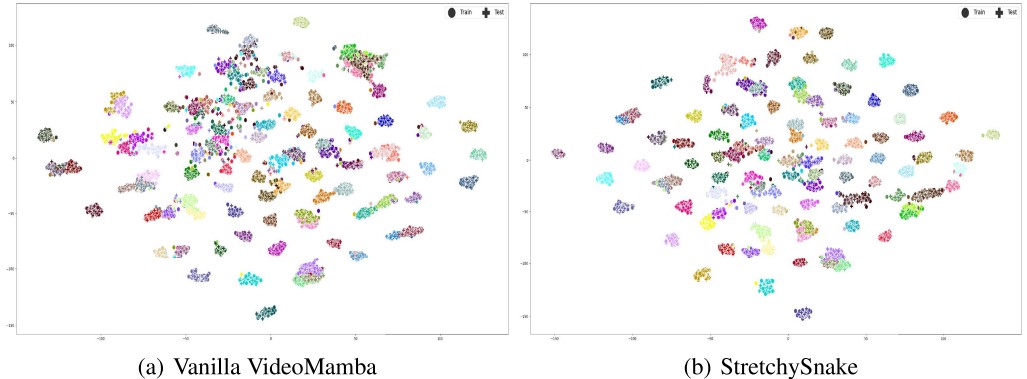

(a) Vanilla VideoMamba                                    (b) StretchySnake

Figure A2: [CLS] token visualization on the UCF-101 dataset at $H = W = 192$ pixels. Each color denotes one class (with some redundancy due to the high number of classes).

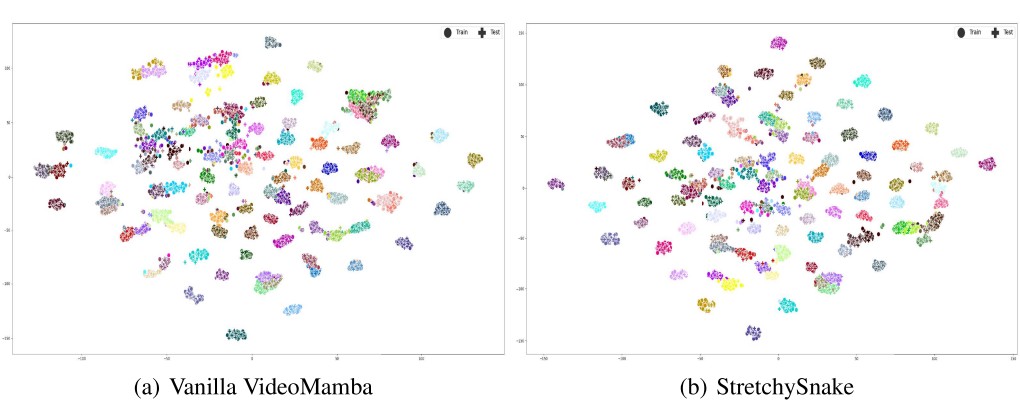

(a) Vanilla VideoMamba        (b) StretchySnake

Figure A3: [CLS] token visualization on the UCF-101 dataset at $H = W = 224$ pixels. Each color denotes one class (with some redundancy due to the high number of classes).

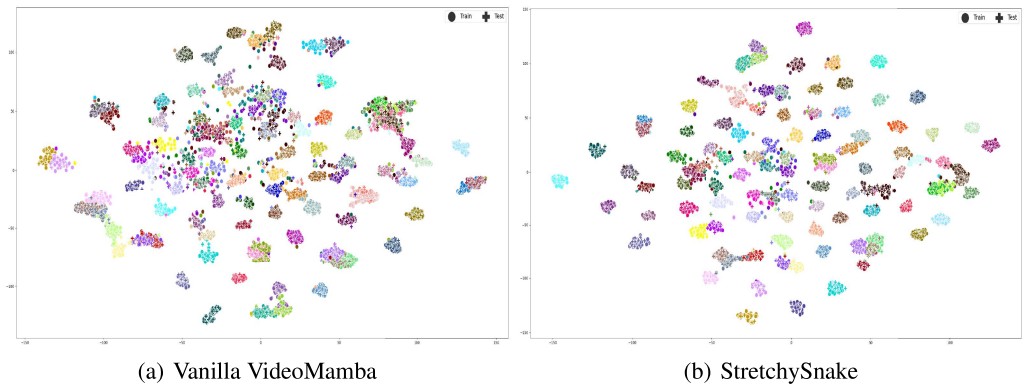

(a) Vanilla VideoMamba        (b) StretchySnake

Figure A4: [CLS] token visualization on the UCF-101 dataset at $H = W = 448$ pixels. Each color denotes one class (with some redundancy due to the high number of classes).

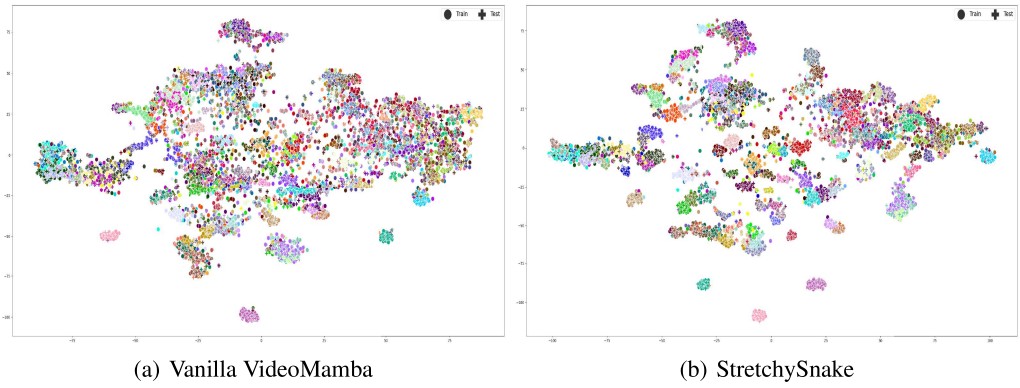

(a) Vanilla VideoMamba        (b) StretchySnake

Figure A5: [CLS] token visualization on the COIN dataset at $H = W = 112$ pixels. Each color denotes one class (with some redundancy due to the high number of classes).

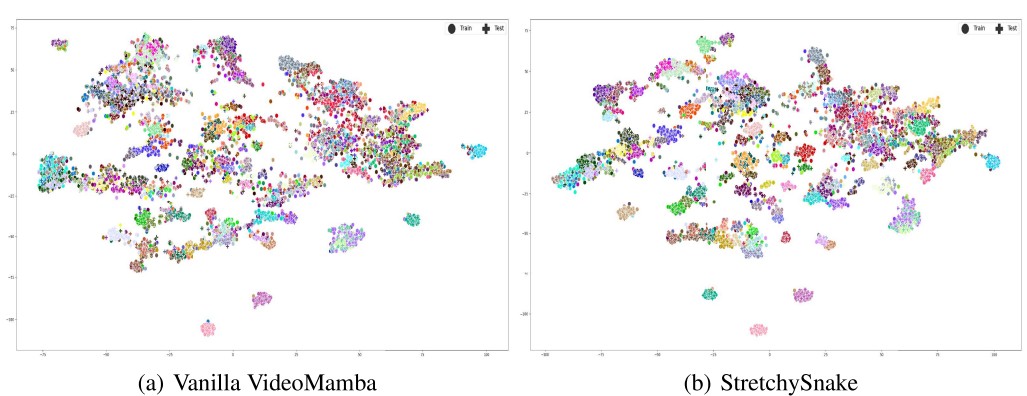

(a) Vanilla VideoMamba          (b) StretchySnake

Figure A6: [CLS] token visualization on the COIN dataset at $H = W = 192$ pixels. Each color denotes one class (with some redundancy due to the high number of classes).

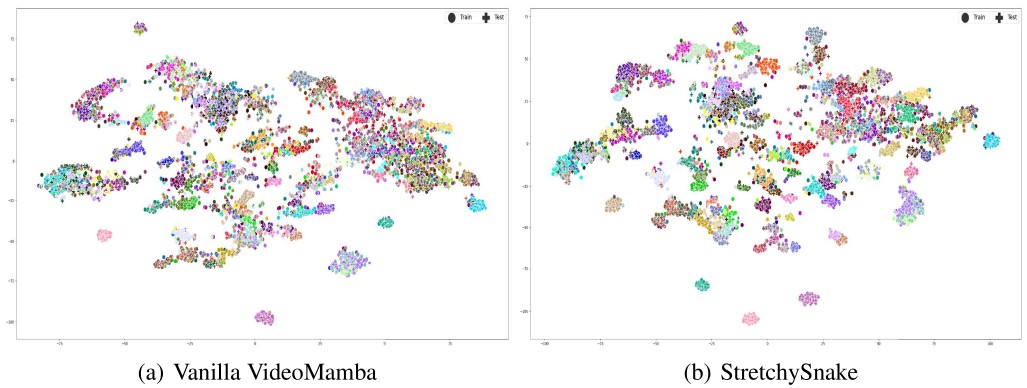

(a) Vanilla VideoMamba          (b) StretchySnake

Figure A7: [CLS] token visualization on the COIN dataset at $H = W = 224$ pixels. Each color denotes one class (with some redundancy due to the high number of classes).

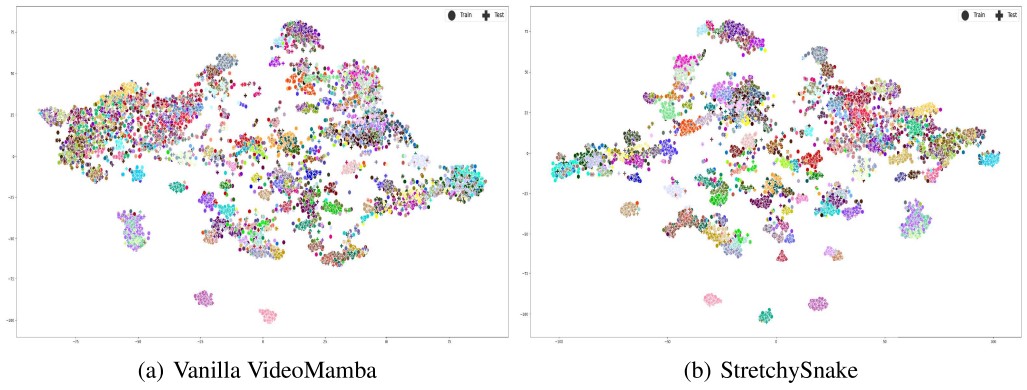

(a) Vanilla VideoMamba          (b) StretchySnake

Figure A8: [CLS] token visualization on the COIN dataset at $H = W = 448$ pixels. Each color denotes one class (with some redundancy due to the high number of classes).

### A.5.2 PATCH ACTIVATION MAPS

In this section, we show additional patch activation maps at different spatial resolutions during video retrieval across the HMDB51 (Figs. A9-A12), COIN (Figs. A13-A16), Breakfast (Figs. A17-A20), and UCF-101 (Figs. A21-A24) datasets. Each graph best viewed with zoom to see finer details. It is important to note that StretchySnake and vanilla VideoMamba are only trained on the Kinetics-400 dataset, so all visualizations are on completely unseen data. We sample every other frame in the interest of space, and remove black frames which are present in some videos of the COIN dataset.

In the HMDB51 activation maps, StretchySnake exhibits impressive abilities such as tracking and activating on faces, even when they change (left) and on complex and fast moving objects (middle, right). This behavior tracks across all spatial resolutions, whereas vanilla VideoMamba struggles to activate on the correct region at lower resolutions (Figs. A9 and A10) while having difficulty *focusing* on the correct region at higher resolutions (Figs. A11 and A12). Similar behavior is seen on the COIN visualizations, with StretchySnake correctly tracking faces and objects whereas vanilla VideoMamba has either uniformly low activations at low resolutions (Figs. A13 and A14) or random activations at higher resolutions (Figs. A15 and A16). Furthermore, the Breakfast activation maps (Figs. A17-A20) highlight how StretchySnake activates on pertinent items for action recognition (like the coffee mug in the left and middle examples, and the eggs and pan in the right example). Lastly, the UCF-101 visualizations simply expound on the patch activation maps shown in the main paper at different spatial resolutions.

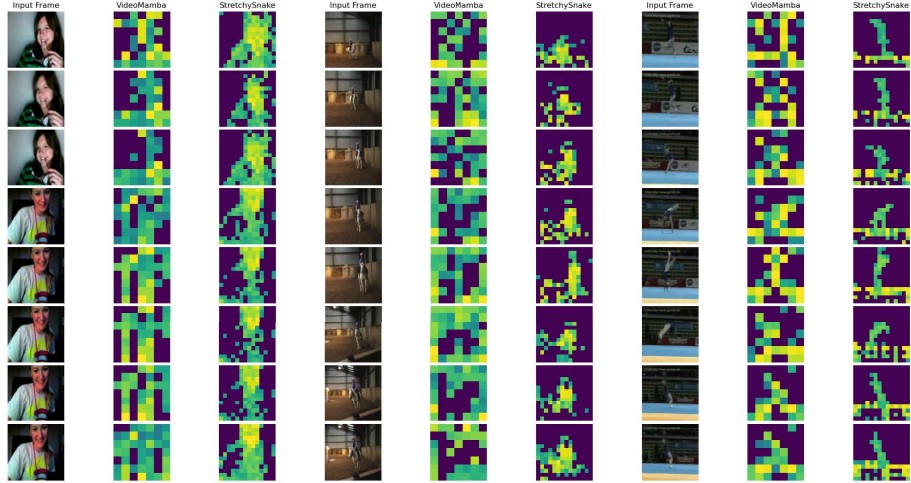

Figure A9: Patch activation map on the HMDB51 dataset at $H = W = 112$ pixels.

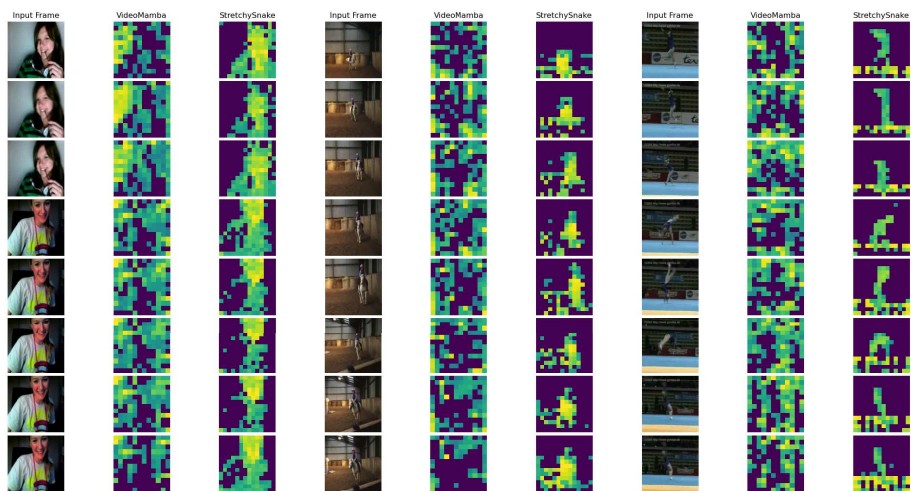

Figure A10: Patch activation map on the HMDB51 dataset at $H = W = 192$ pixels.

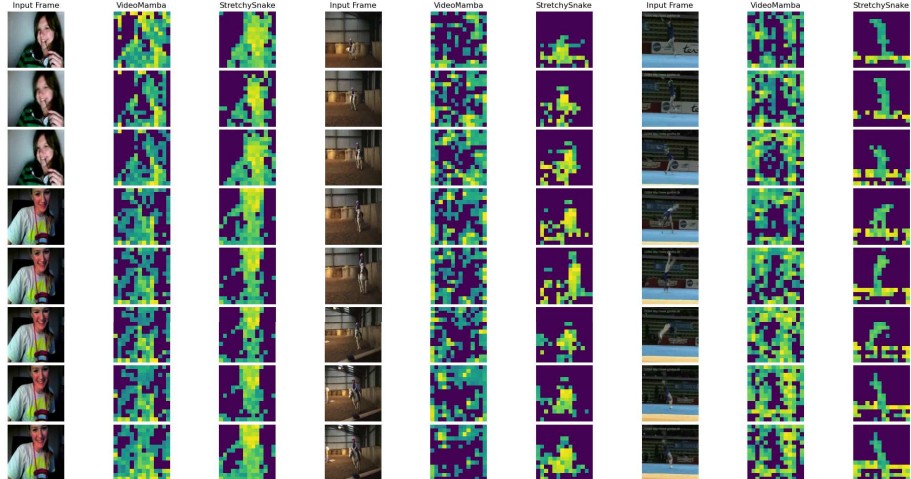

Figure A11: Patch activation map on the HMDB51 dataset at $H = W = 224$ pixels.

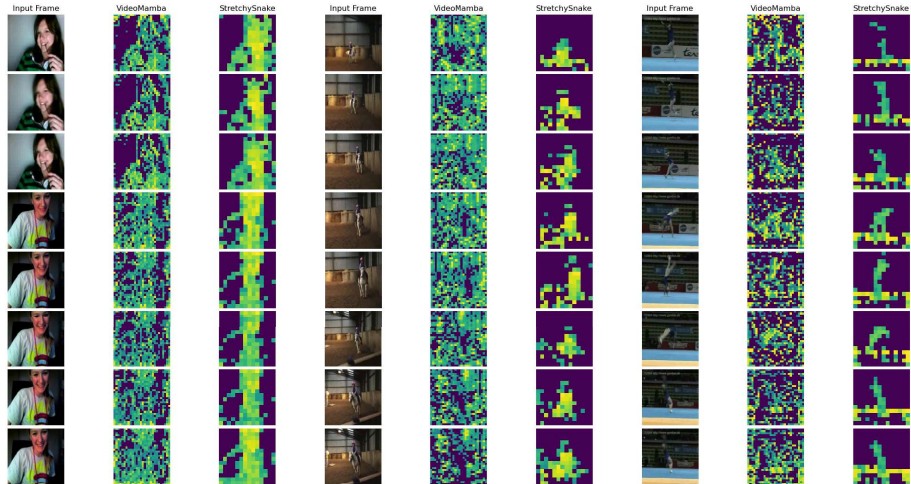

Figure A12: Patch activation map on the HMDB51 dataset at $H = W = 448$ pixels.

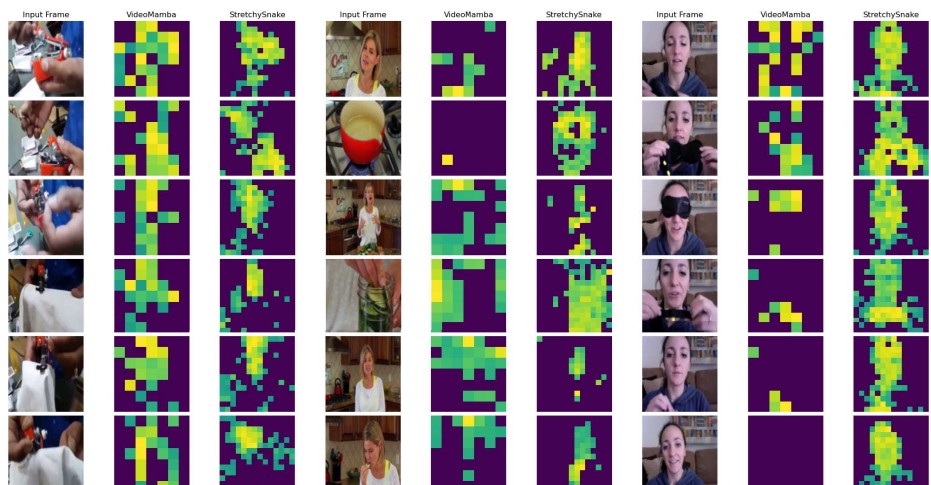

Figure A13: Patch activation map on the COIN dataset at $H = W = 112$ pixels.

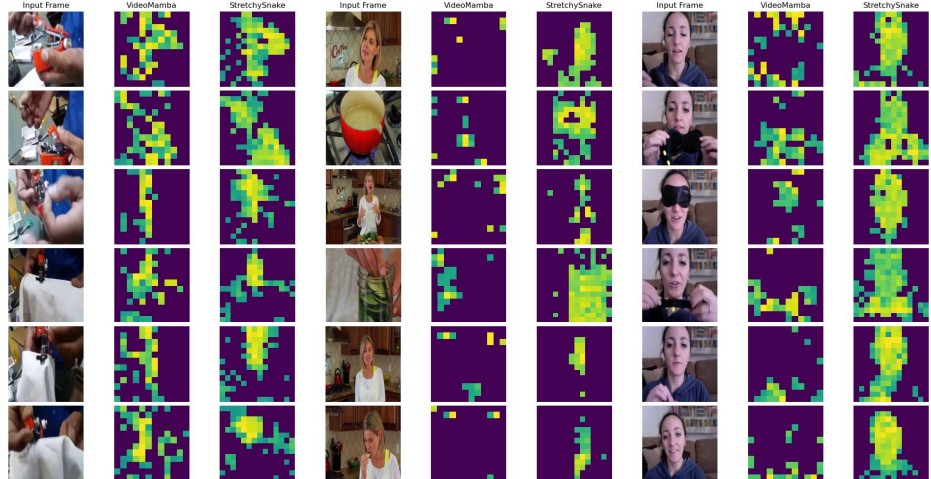

Figure A14: Patch activation map on the COIN dataset at $H = W = 192$ pixels.

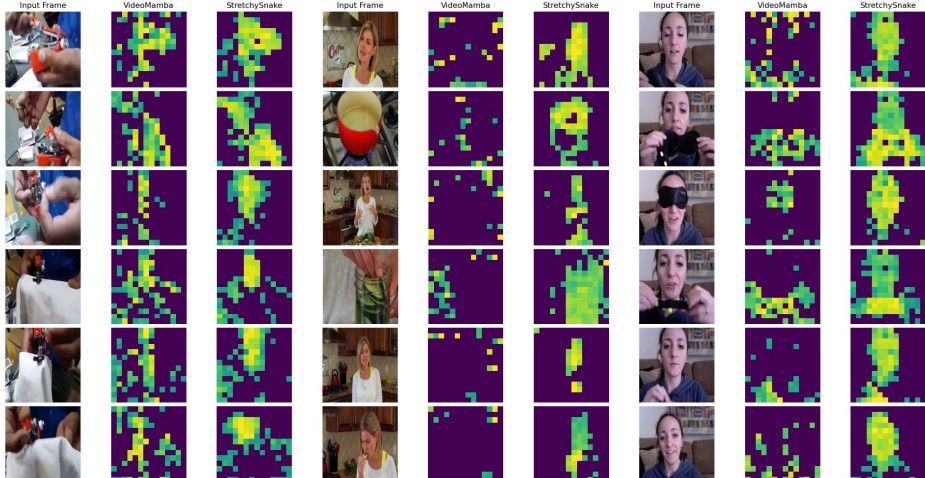

Figure A15: Patch activation map on the COIN dataset at $H = W = 224$ pixels.

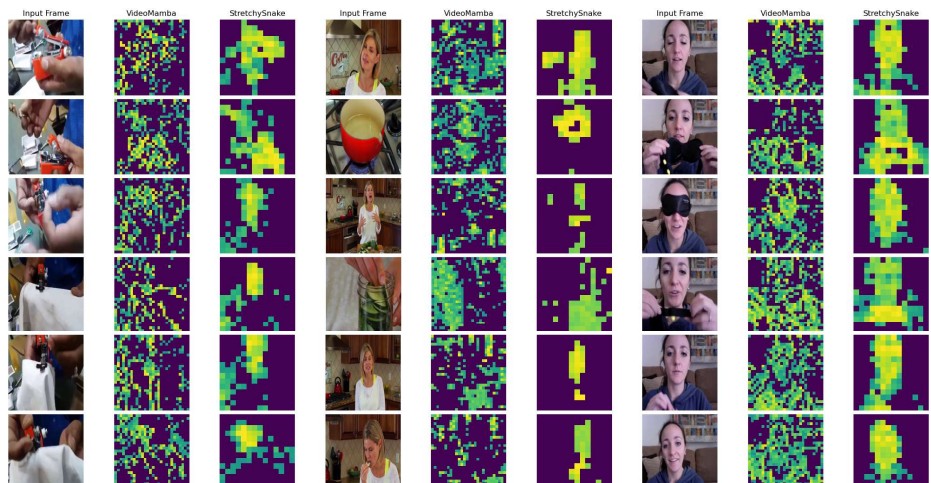

Figure A16: Patch activation map on the COIN dataset at $H = W = 448$ pixels.

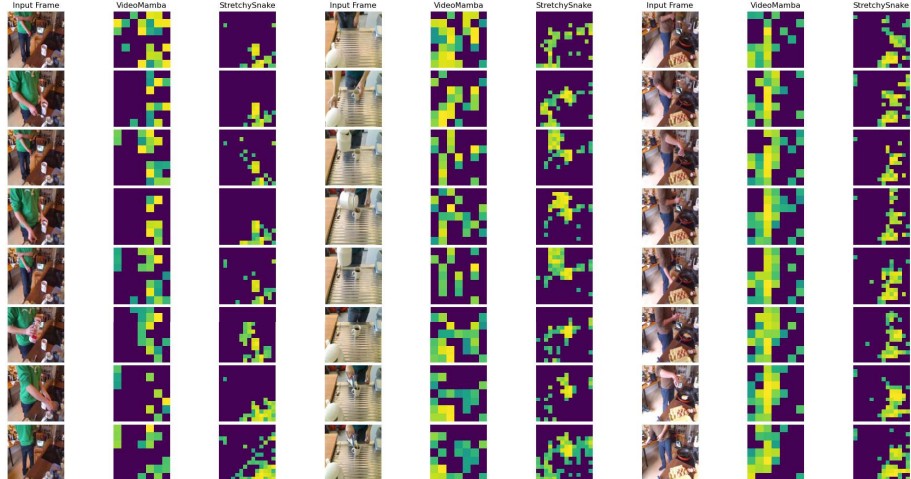

Figure A17: Patch activation map on the Breakfast dataset at $H = W = 112$ pixels.

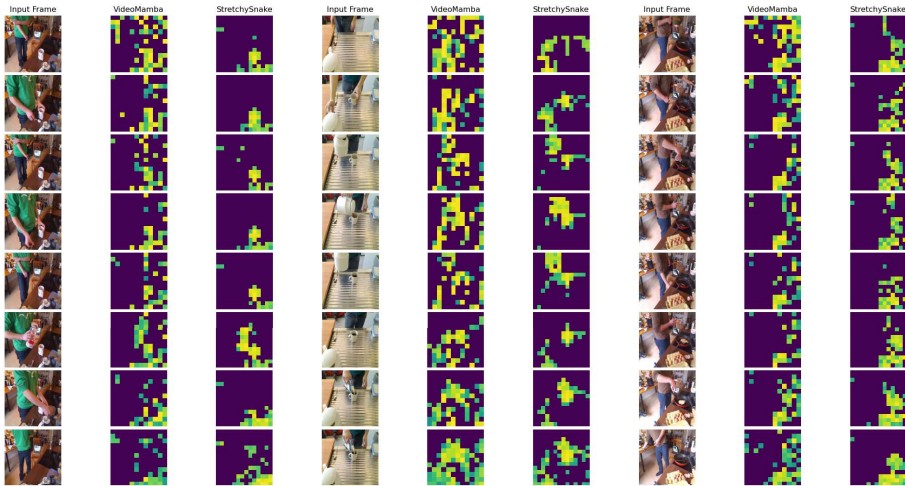

Figure A18: Patch activation map on the Breakfast dataset at $H = W = 192$ pixels.

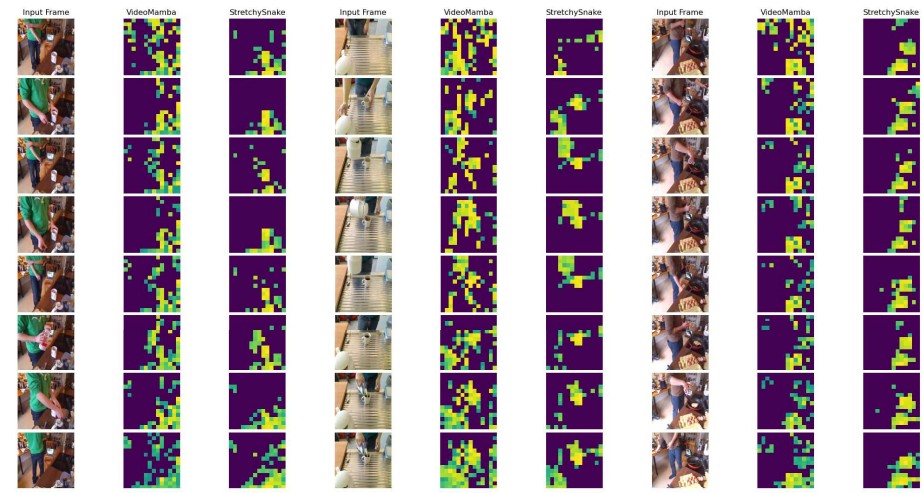

Figure A19: Patch activation map on the Breakfast dataset at $H = W = 224$ pixels.

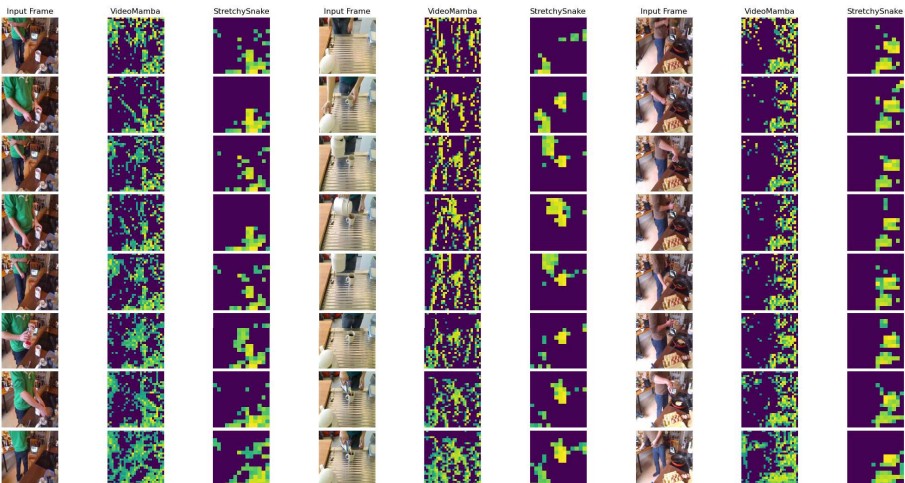

Figure A20: Patch activation map on the Breakfast dataset at $H = W = 448$ pixels.

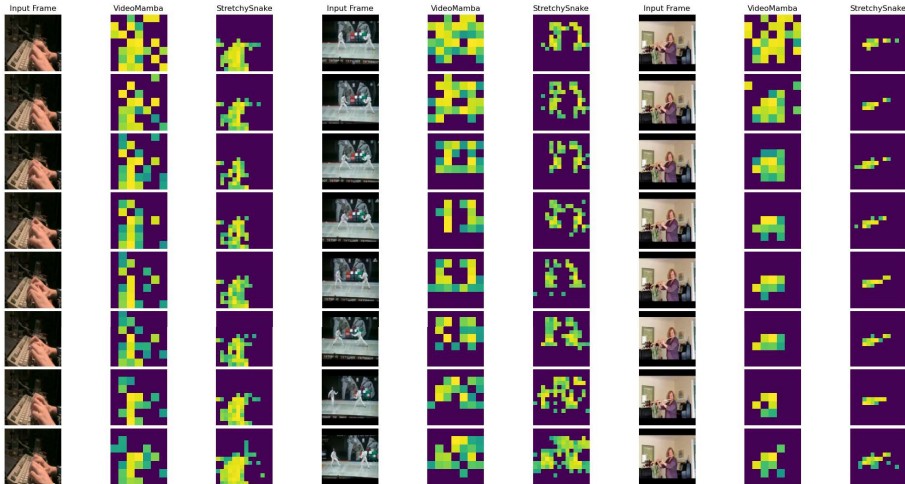

Figure A21: Patch activation map on the UCF-101 dataset at $H = W = 112$ pixels.

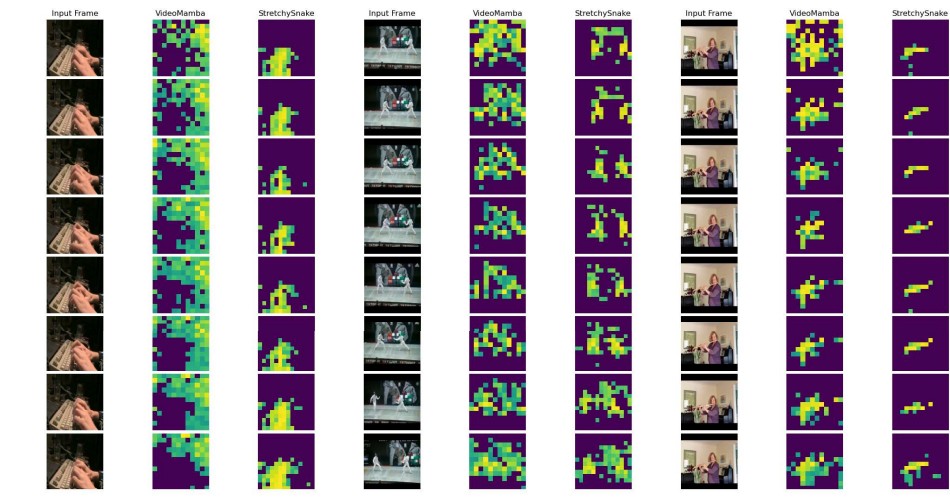

Figure A22: Patch activation map on the UCF-101 dataset at $H = W = 192$ pixels.

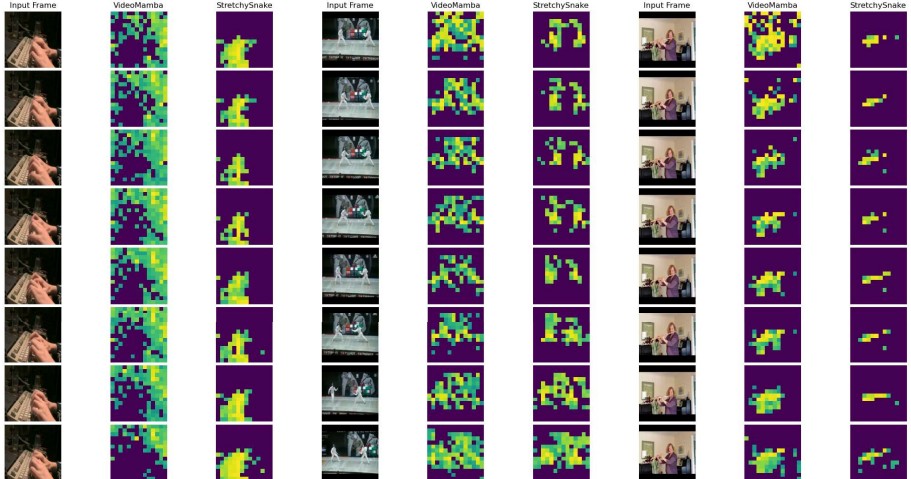

Figure A23: Patch activation map on the UCF-101 dataset at $H = W = 224$ pixels.

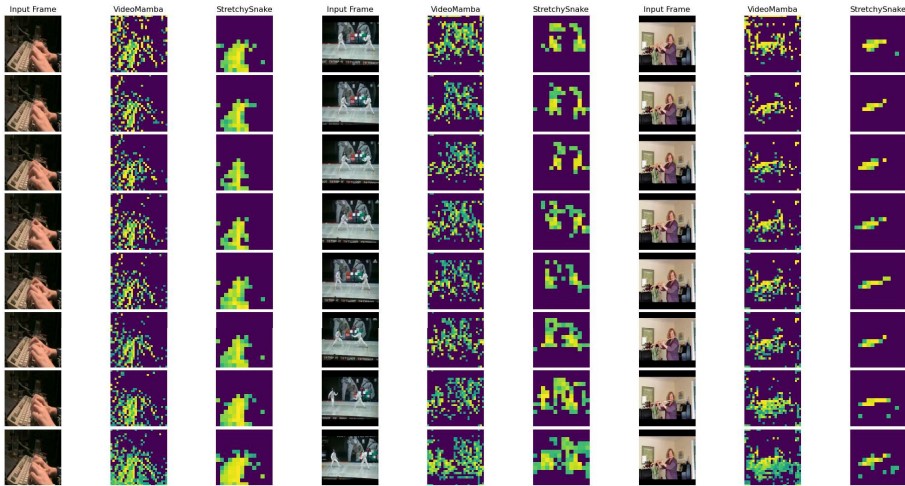

Figure A24: Patch activation map on the UCF-101 dataset at $H = W = 448$ pixels.