# OpenReview forum: "StretchySnake: Flexible VideoMamba for Short and Long-Form Action Recognition"
_ICLR.cc/2025/Conference — Submitted to ICLR 2025_

### Official Review · Reviewer_oh1A · 2024-11-02

**Soundness:** 3
**Presentation:** 3
**Contribution:** 3
**Rating:** 6
**Confidence:** 4

**Summary:**

This paper aims to train a flexible VideoMamba architecture, which can handle short and long-term video understanding by adapting different video resolution and temporal sampling rate. Accordingly, the paper proposes a simple yet effective stragety based on dynamically interpolating model weights. The authors conduct experiments on four short or long-term video datasets, namely UCF101, HMDB51, COIN and Breakfast, and the results demonstrate that the proposed training method can improve VideoMamba effectively.

**Strengths:**

1. The authors propose a simple yet effective method to adapt different video inputs, which seems to generalize to other domains.
2. The proposed model can handle both short-term and long-term videos, which advances the applications of VideoMamba.

**Weaknesses:**

1. The authors conduct experiments on UCF101, HMDB51, COIN and Breakfast datasets, which are relatively small datasets. Can you provide results on relative larger datasets, e.g., Kinetics400, something-something-v2.
2. As stated in Line 299~300, the authors verify the effectiveness of the proposed model mainly based on transfer learning experiments. How about the performance in vanilla fully-supervised setting?
3. Can the proposed training recipe benefits downstream video understanding tasks, e.g., weakly-supervised action localization?

If the authors can address my concerns above, I am willing to raise my rating.

**Questions:**

1. In Line 347~348, the authors conclude that the best type of st-flexibility modeling is static-tokens, according to the empirical analysis. Can you explain the reason in model design or learning? I think any insights you provide will benefit the society.
2. As stated in Line 316~317, on Kinetics-400, you flexibly train the proposed model for 12 epochs, while train the vanilla VideoMamba for 50 epochs. Why? Is it not fair? How do the two models perform on Kinetics-400?

---

> ### Author Response · Authors · 2024-11-21
>
> > "The authors conduct experiments on UCF101, HMDB51, COIN and Breakfast datasets, which are relatively small datasets. Can you provide results on relative larger datasets, e.g., Kinetics400, something-something-v2."
>
> We address in a block at the top titled "Shared Comments". Kindly refer to Q1 in the "Shared Comments" block.
>
> > "As stated in Line 299~300, the authors verify the effectiveness of the proposed model mainly based on transfer learning experiments. How about the performance in vanilla fully-supervised setting?"
>
> Kindly refer to Q2 in the "Shared Comments" block.
>
> > "Can the proposed training recipe benefits downstream video understanding tasks, e.g., weakly-supervised action localization?"
>
> Kindly refer to Q3 in the "Shared Comments" block.
>
> > "The authors conclude that the best type of st-flexibility modeling is static-tokens, according to the empirical analysis. Can you explain the reason in model design or learning? I think any insights you provide will benefit the society."
>
> Thank you for the interesting question! The original VideoMamba paper examined 4 different types of scanning methods determining in which order the temporal or spatial dimension is scanned. This is similar to the factorized attention idea seen in ViViT [1] and/or TimeSformer [2] (which are cited in the main paper as well). The VideoMamba authors found that "spatial-first, then temporal" scanning performed the best. Moreover, we found it interesting that our "flex-all" method was not the best performing method, as we hypothesized that the method with the most flexibility would learn the best representations. Our analysis found that the “static-tokens“ approach seems to work the best, and we believe it is since it is the only flexible method that keeps the number of spatial tokens fixed. Since VideoMamba selectively scans along the spatial dimension first, it must compress only the salient spatial information (the main function of SSMs) into a fixed number of spatial tokens. Then, since the temporal scan is second and SSMs prioritize newer context, it is easier to "flex" or interpolate across the temporal dimension to choose only the salient temporal information, while the older spatial information is always of a fixed size. We hypothesize that attempting to also flex the number of spatial tokens during training makes it more difficult for VideoMamba's selective scan to retain the "older context" spatial information when the allotted tokens are constantly changing, and especially when temporal selective scanning introduces newer, higher priority context. We believe this further adds to the contribution of our work, as our different versions of flexibility could be explored in the future for different visual tasks - maybe the “static-tokens” approach is not the best for tasks that require different visual focus than action recognition, such as object tracking or video segmentation.
>
> > "As stated in Line 316~317, on Kinetics-400, you flexibly train the proposed model for 12 epochs, while train the vanilla VideoMamba for 50 epochs. Why? Is it not fair? How do the two models perform on Kinetics-400?"
>
>
> Sorry for the confusion, we added this point for full transparency but more importantly to highlight that flexible training leads to much faster convergence. We saw StretchySnake's validation accuracy on Kinetics start to saturate around the 12th epoch, while it took vanilla VideoMamba 50 epochs to reach peak accuracy. If anything, it is a more unfair comparison for StretchySnake since we train for less epochs than vanilla VideoMamba, yet we still report massive performance gains in the paper. We report the accuracies of StretchySnake and vanilla VideoMamba on Kinetics-400 and Kinetics-700 in Q1 in the "Shared Comments" block.
>
>
> 1. Arnab, A., Dehghani, M., Heigold, G., Sun, C., Lučić, M., & Schmid, C. (2021). Vivit: A video vision transformer. In Proceedings of the IEEE/CVF international conference on computer vision (pp. 6836-6846).
>
> 2. Bertasius, G., Wang, H., & Torresani, L. (2021, July). Is space-time attention all you need for video understanding?. In _ICML_ (Vol. 2, No. 3, p. 4).

---

> > ### Comment · Reviewer_oh1A · 2024-11-29
> >
> > The authors don't provide the results of K400/SSv2 action recognition, they provide the K400 video retrieval results. Although these experiments may have high computation cost, they are necessary for a video architecture paper. Thus, I maintain my rating.

---

> ### Author Response · Authors · 2024-12-01
>
> In order to further exhibit the benefit of training with st-flexibility, we provide video retrieval, linear probing, and finetuning results on SSV2. We also provide evaluation results wherein we take the fine-tuned model and perform inference using a different number of frames, and to specifically address the reviewer's concerns, additional classification results on K400 are given. It is important to note that since StretchySnake and VideoMamba are pre-trained on K400, video retrieval and video classification results are nearly identical, hence why we initially omitted it.
>
>
>
> ## Results on SmthSmthV2
> | Model | Video Retrieval | Linear Probing | Finetune (eval @ 16 frames) | Finetune (eval @ 64 frames)
> | :------: | :------: | :------: | :---: | :---: |
> | VideoMamba | $9.8$  | $14.3$ | $64.5$ | $63.8$ |
> | StretchySnake | $11.7$ (+$1.9$) | $17.9$ (+$3.6$) | $69.8$ (+$5.3$) | $70.5$ (+$6.7$)|
>
> ## Results on K400
> | Model | Video Retrieval | Classification
> | :------: | :------: | :------: |
> | VideoMamba | $78.0$ | $78.6$
> | StretchySnake | $78.2$ (+$0.2$) | $79.0$ (+$0.4$) |
>
> Again, we see that StretchySnake outperforms VideoMamba in every facet, especially when generalizing to unseen data.  Furthermore, we show in the final two columns of the SSV2 table that the fine-tuned StretchySnake achieves higher accuracy when using $64$ frames instead of $16$ during inference, whereas VideoMamba is not able to exploit the additional temporal information. This is a result of our st-flexible training method and further exhibits the key contributions of our work.
>
> **As we discussed in part 2 of the "Shared Comments" block, we have now shown results on short-form, coarse-grained action recognition (UCF101, HMDB51), long-form, coarse-grained action recognition (COIN, Breakfast), large-scale action recognition (K400, K700), and fine-grained action recognition (Diving48, SSV2)**. We thank the reviewers for their suggestions as we believe this further exhibits the strengths of StretchySnake, and we believe that the many evaluation protocols we use across all of these datasets are more than sufficient to prove the benefit of st-flexbile training for SSMs. In addition to the qualitative results shown in the paper, we hope the reviewers agree that our method has been rigorously tested across all major benchmark datasets and thus could provide benefit to the greater community as a whole as well. We believe all of your original concerns have now been addressed, and please note we will add all of these experiments in the appendix in the camera-ready version upon acceptance.

---

> > ### Comment · Reviewer_oh1A · 2024-12-02
> >
> > Thank you for your additional results. These results look good.
> >
> > I wonder the detailed experiment settings of these results, e.g.,  epoch number, pretraining setting, parameter scale, dataset split. Please clarify these in an organized way as detailed as you can. If your clarification convinces me, I am willing to increase my rating.

---

> ### Author Response · Authors · 2024-12-02
>
> Sure, we would be happy to provide those details! We follow the same exact fine-tuning parameters that VideoMamba uses for fine-tuning:
> - $\\#$ of Training Epochs: 30
> - Pretraining setting: Both models are pre-trained on K400 (StretchySnake is pre-trained using st-flexible training as discussed in our paper), then fine-tuned on SSV2
> - Learning Rate Schedule: cosine decay
> - Warmup Epochs: 5
> - Learning Rate: $1.0 \times 10^{-4}$
> - Dataset Split: We follow the standard dataset split of SSV2 as is provided with the dataset
> - Optimizer: AdamW ($\beta_1 = 0.9, \beta_2 = 0.999$)
> - Weight Decay: 0.05
>
> These same exact configurations are given in the VideoMamba codebase (which we built StretchySnake upon) and can be found in Table V of the original VideoMamba paper (with additional details). This is the same configuration we used for every fine-tuning experiment across every dataset reported in the paper to provide the fairest comparisons with VideoMamba. We hope this answers all of your questions!

---

> > ### Comment · Reviewer_oh1A · 2024-12-02
> >
> > How about the K400 experiments?

---

> ### Author Response · Authors · 2024-12-02
>
> The K400 experiments do not require any fine-tuning: the models are pre-trained in a supervised fashion with the provided K400 ground truth labels. We directly took these models and performed inference on video retrieval and video classification as requested. We provide the pre-training training details in Section $5.1$ of the paper (Lines $314$-$336$) and again they are identical to the pre-training configuration used in the VideoMamba paper, obviously besides st-flexible training (which requires less training epochs, as addressed in our first response to you in the last paragraph). We hope this helps in clarifying everything!

---

> > ### Comment · Reviewer_oh1A · 2024-12-02
> >
> > Thank you for your detailed response. I will increase my rating to 6.

---

### Official Review · Reviewer_TETD · 2024-11-03

**Soundness:** 3
**Presentation:** 3
**Contribution:** 3
**Rating:** 6
**Confidence:** 3

**Summary:**

This paper introduces StretchySnake, a training method for video state space models (SSMs) that enhances spatio-temporal flexibility. The approach involves randomly varying the spatial and temporal resolutions of videos during training, allowing a single model to adapt to different input sizes without retraining. The authors evaluate five different spatio-temporal flexible training methods and find that "static tokens" perform optimally. StretchySnake, their best-performing model, significantly outperforms the standard VideoMamba across various action recognition datasets,

**Strengths:**

In general, the proposed method is somehow simple yet effective. The proposed method explores different settings of making the fixed resolution VideoMamba flexible to dynamic spatial and temporal resolution. Also, the proposed method performs extensive experiments on both short and long action recognition datasets to verify the effectiveness of the proposed method. Various visualization also makes the results more convincing.

**Weaknesses:**

Besides the strengths part, I still have some concerns about the paper:

1. The author starts to fine-tune the model from a self-supervised trained VideoMamba model. Then the author claimed that the static tokens perform best at flexible training, i.e., different time scales and different spatial resolutions. Will the self-supervised training (VideoMAE with token masking) dominate the downstream pattern and performance? I think solely fine-tuning and line-probing a pre-trained fixed-resolution model is less convincing than flexible training a model from scratch and then applying it in the downstream tasks. The author could perform a from-scratch pipeline, even in a smaller dataset to verify the effectiveness of the proposed method to make the result more convincing.

2. The author could have more datasets to convince the model. For example, the author chooses two appearance-based action recognition dataset like UCF101 and HMDB51, and 2 multi-label datasets like BreakFast and COIN. What if the proposed method are tested on some temporal sensitive datasets like Sth-Sth V2? If the author could verify that, I think the proposed paper can be more convincing.

3. For FLOPs calculation, the author could compare more with test time augmentations (i.e.), with fixed resolution models, but performing more inferences. Applying dynamic resolution to a fixed-resolution model is sometimes unfair here.

**Questions:**

Based on the weaknesses mentioned above, I have the following questions:

1. Is the static token pattern dominated by the upstream self-supervised learning tasks?

2. Will the static-token pattern also performs better on temporal related datasets like Sth-Sth V2?

---

> ### Author Response · Authors · 2024-11-21
>
> > "Will the self-supervised training (VideoMAE with token masking) dominate the downstream pattern and performance? Is the static token pattern dominated by the upstream self-supervised learning tasks?"
>
> Regarding training both vanilla VideoMamba and StretchySnake from scratch, please refer to Q2 in the "Shared Comments" block at the top as this was a common question among two reviewers. As for whether static-tokens only performs the best due to the upstream weights, this is definitely an interesting question. Important to note, the papers [1, 2] that we discuss in the main paper that explore flexible training for images both initialize their models with pre-trained weights and show that further flexible training still vastly improves performance. We felt that first loading VideoMamba's weights would be more practical for this work, as it is unlikely and impractical for any interested readers of VideoMamba or our paper to train VideoMamba again from scratch. This is in addition to the fact that VideoMamba's training is quite sensitive and requires multiple stages (which can be found in their original paper [3], such as first training on images, distilling from a larger model, masked training, etc.), which could also interfere with properly analyzing our proposed training method. However, investigating which of our proposed methods of st-flexibility is optimal for other training objectives/settings could serve as excellent future work (as we discuss with Reviewer oh1A as well).
>
> > The author could have more datasets to convince the model. For example, the author chooses two appearance-based action recognition dataset like UCF101 and HMDB51, and 2 multi-label datasets like BreakFast and COIN. What if the proposed method are tested on some temporal sensitive datasets like Sth-Sth V2? If the author could verify that, I think the proposed paper can be more convincing.
>
> We answer this in the "Shared Comments" section at the top. Please refer to Q1 and Q3 for our detailed response
>
> > For FLOPs calculation, the author could compare more with test time augmentations (i.e.), with fixed resolution models, but performing more inferences. Applying dynamic resolution to a fixed-resolution model is sometimes unfair here.
>
> We are sorry, we are not exactly sure what this is referring to. If referring to Figure 1, this figure is meant to compare StretchySnake's video retrieval results at the lowest spatial and temporal resolution (H=W=96 pixels, T=8 frames) against the $\textit{highest}$ accuracies attained by vanilla VideoMamba. Even if we cherry-pick the highest accuracies for vanilla VideoMamba across every spatial and temporal resolution for each dataset from Table 1, Figure 1 exhibits how StretchySnake $\textit{still}$ outperforms vanilla VideoMamba while also requiring much less compute during inference. If we were to use vanilla VideoMamba's default configuration (H=W=224 pixels, T=16 frames) which would reduce the GFLOPs and inference time, the accuracies for VideoMamba in the figure would drop, further exhibiting StretchySnake's superior performance. We would be happy to discuss further if this was not the main concern of the reviewer from this comment.
>
> > "Will the static-token pattern also performs better on temporal related datasets like Sth-Sth V2?"
>
> Please refer to Q3 in the "Shared Comments" section at the top, where we evaluate our method on the highly temporally-related action recognition dataset Diving-48.
>
> 1. Beyer, L., Izmailov, P., Kolesnikov, A., Caron, M., Kornblith, S., Zhai, X., ... & Pavetic, F. (2023). Flexivit: One model for all patch sizes. In _Proceedings of the IEEE/CVF Conference on Computer Vision and Pattern Recognition_ (pp. 14496-14506).
> 2. Tian, R., Wu, Z., Dai, Q., Hu, H., Qiao, Y., & Jiang, Y. G. (2023). Resformer: Scaling vits with multi-resolution training. In _Proceedings of the IEEE/CVF Conference on Computer Vision and Pattern Recognition_ (pp. 22721-22731).
> 3. Li, K., Li, X., Wang, Y., He, Y., Wang, Y., Wang, L., & Qiao, Y. (2025). Videomamba: State space model for efficient video understanding. In _European Conference on Computer Vision_ (pp. 237-255). Springer, Cham.

---

> > ### Comment · Reviewer_TETD · 2024-11-26
> > **Response to the rebuttal**
> >
> > Thank the author for providing the rebuttal. I'm glad that most of my concerns are addressed. I will keep my current score leaning to accept. However, the experiments in the rebuttal still focus on standard recognition/retrieval tasks. Also, I don't think diving48 is a good dataset (maybe sth-sth v2 is better), since a simple method (without temporal module design) can also perform well on diving48.

---

> > > ### Author Response · Authors · 2024-12-01
> > >
> > > Thank you for your suggestion! Please see our response to Reviewer oh1A where we provide additional results on SmthSmthV2.

---

### Official Review · Reviewer_gT1y · 2024-11-03

**Soundness:** 2
**Presentation:** 1
**Contribution:** 1
**Rating:** 1
**Confidence:** 4

**Summary:**

To develop spatio-temporal feature representations that are robust across varying video lengths, the paper exposes pretrained VideoMamba models to random spatio-temporal input resolutions via resizing operations. The proposed method is evaluated on the UCF101, HMDB51, Breakfast, and COIN datasets. Ablation studies are conducted for different spatio-temporal resolutions, and visual analysis for video retrieval is also provided.

**Strengths:**

- The implementation details are sufficiently thorough, enhancing the reproducibility and reliability of this work
- The visual analysis provided is clear and convincing.

**Weaknesses:**

- The intended purpose of Figure 2 is unclear. For instance, the spatial positional embeddings are obscured by other elements, and the temporal positional embedding, defined as $\textbf{ξ}^t$ with shape $1 \times S \times D$ on line 207, is illustrated as having only shape $D$.  On line 194, $S=H \times W \ (p \times p)$, but $S = 128/16=8$, which is very confused.

- There is a mismatch between Fig.(2) and its caption, where the figure merely depicts a standard tokenization process for videos, while the caption emphasizes 'flexibility,' which is not effectively conveyed in this diagram.

- The paper uses mathematical symbols inappropriately and inconsistently with established conventions, making it challenging for readers to fully understand the paper.

- There is a lack of comparisons with state-of-the-art (SOTA) methods on action recognition tasks. Additionally, StretchySnake’s performance is surpassed by SOTA methods, such as VideoMAE, which achieves 99.6% top-1 accuracy on UCF101, significantly higher than StretchySnake’s 96.5%.

- The four datasets used in the paper are small-scale. No results on larger scale datasets, such as Kinetics400/600 and Something-Something are provided in this paper.

- Overall, the concept of "st-flexibility" amounts to a straightforward spatio-temporal interpolation of the input, limiting the novelty of the idea and offering minimal contribution to the community.

**Questions:**

Please refer to the Weaknesses.

---

> ### Author Response · Authors · 2024-11-21
> **Official Comment by Authors (1/2)**
>
> >Quality and Presentation of Figure 2
>
> We agree with the reviewer on their first two points, and as such, have made several edits to Figure $2$ in the updated pdf to make it more clear and to better highlight our proposed "flexible" method of training. The new figure can be viewed in the updated pdf (highlighted in blue), and we describe the major changes below.
>
>
> Firstly, the shape of $\xi^t$ is defined by the concatenation of every patch embedding ($e^s \in \mathbb{R}^D$) for a single frame. Thus, if there are $S=14$ spatial tokens for a single frame, the concatenation of them results in a tensor of shape $14 \times D$. We add an additional dimension for later concatenation across all frames (temporal dimension), hence the resulting shape of $1 \times S \times D$. The larger rectangle shape of $\xi^t$ in the original figure may not have fully conveyed this and thus has been changed.
>
> Secondly, we acknowledge the inconsistency in the representation of spatial and temporal positional embeddings, and we have addressed this. Both are now clearly represented for improved clarity.
>
> Lastly, we have added more detail about how exactly we implement flexibility in our proposed method. Kindly refer to the updated figure in the PDF.
>
> > On line 194, $S= H \times W / (p \times p)$, but, $S = 128/16 = 8$, which is very confused.
>
>
> You are correct this is a typo, thank you for pointing that out and sorry for the confusion. $S$ is meant to denote the number of spatial tokens produced after "patchification" and computing the embeddings of each patch (Lines $197-201$). As an example and to provide additional clarity, if the image size is $224^2$ and the patch size is 16, then the number of resulting spatial tokens is represented as $S = \sqrt{\frac{H}{ps} \times \frac{W}{ps}}= \sqrt{\frac{224}{16} \times \frac{224}{16}} = \sqrt{\frac{224\times 224}{16 \times 16}} = 14$. This text and all subsequent relevant text has been fixed in the updated pdf (highlighted in blue).
>
>
> > The paper uses mathematical symbols inappropriately and inconsistently with established conventions, making it challenging for readers to fully understand the paper.
>
>
> We are sorry for the confusion stemming from our choice of mathematical notation. We appreciate the reviewer's comment and decided to change whatever notation we can to match VideoMamba, as that is the most relevant paper to our work and to any potential future readers. This includes changes such as changing spatial and temporal position embeddings to $\mathbf{p}\_{s}$ and $\mathbf{p}\_{t}$, respectively, changing the number of spatial tokens from $S$ to $L$, and referring to patch size as $p$, among other changes that can be found highlighted in blue in the updated pdf. However, we found it difficult to find an established convention across relevant literature. For example, we define our patch embeddings as $e^s$, but ViT [1] uses $E$, ResFormer [2] uses $x^{img}$, FlexiViT [3] uses $e_i$, and VideoMamba [4] uses $X^p$. Another example is how we define spatial positional embeddings as $\xi^s$, but ViT [1] uses $E_{pos}$, ResFormer [2] uses $p$, FlexiViT [3] uses $\pi_i$, and VideoMamba [4] uses $\mathbf{p}\_{s}$. If there was a specific convention the reviewer had in mind, we would be happy to accommodate that as well.
>
>
> > There is a lack of comparisons with state-of-the-art (SOTA) methods on action recognition tasks. Additionally, StretchySnake’s performance is surpassed by SOTA methods, such as VideoMAE, which achieves 99.6% top-1 accuracy on UCF101, significantly higher than StretchySnake’s 96.5%.
>
>
> The $99.6\\%$ that the reviewer quotes comes from VideoMAEv2 [5], which was pre-trained on a much larger dataset containing 2M videos, far beyond Kinetics-400 (which contains ~240K training videos). Thus, it is not a fair comparison to our method. Besides, we beat every single baseline listed in the table where VideoMaeV2 [5] reports their results on UCF-101 (including the original VideoMAE which was only trained on Kinetics-400).
>
> Furthermore, the $96.5\\%$ that we report on UCF-101 comes from Table 2, where we exhibit that StretchySnake has learned better quality representations than vanilla VideoMamba due to flexible training (Lines 372-377, 409-410). The purpose of this table is to directly compare StretchySnake and vanilla VideoMamba, not other SOTA models. Take for example Table 3 in our paper, where we consider the listed methods as fair comparisons as they were only trained on Kinetics-400. We even include some methods trained on extra data or multiple modalities in gray. Due to the relative infancy of SSMs in visual tasks, papers which apply SSMs to videos (such as VideoMamba) aim to show that SSMs achieve competitive results against transformers and are viable option moving forward, as opposed to trying to achieve SOTA results.

---

> ### Author Response · Authors · 2024-11-21
> **Official Comment by Authors (2/2)**
>
> > The four datasets used in the paper are small-scale. No results on larger scale datasets, such as Kinetics400/600 and Something-Something are provided in this paper.
>
> We have included additional experiments on larger-scale datasets in the "Shared Comments" section at the top, since this was a common concern. Please refer to Q1 in that section which hopefully address your concern.
>
>
> > Overall, the concept of "st-flexibility" amounts to a straightforward spatio-temporal interpolation of the input, limiting the novelty of the idea and offering minimal contribution to the community.
>
> We agree with the reviewer that st-flexibility itself is a simple concept, but as Reviewer TETD and Reviewer oh1A point out, this can serve as a strength as it is intuitive yet extremely effective. This also allows for easier implementation in other models/training pipelines than just VideoMamba due to its high compatibility.
>
>
>
> We believe our work does offer a significant contribution to the community as a whole, as we not only introduce st-flexibility training as a concept, but analyze multiple possible flexible variants, across different temporal and spatial resolutions, AND across different datasets (Sec. 4.2, Sec. 5.2, and Fig. 3). As previously mentioned, SSMs have shown great promise in the computer vision field but are yet to truly outperform transformers in a general capacity. We show that our method of flexibly training video-based SSMs is highly effective at learning better representations for video retrieval (Tables 1 and 3), and even generally improve the representations of the model itself for downstream tasks (Table 2). In addition to the qualitative visualizations and ablations in the supplementary, we believe these findings could be very valuable to the community in fostering the increased use of SSMs in computer vision.
>
>
> 1.  Dosovitskiy, A. (2020). An image is worth 16x16 words: Transformers for image recognition at scale. arXiv preprint arXiv:2010.11929.
>
> 2.  Tian, R., Wu, Z., Dai, Q., Hu, H., Qiao, Y., & Jiang, Y. G. (2023). Resformer: Scaling vits with multi-resolution training. In Proceedings of the IEEE/CVF Conference on Computer Vision and Pattern Recognition (pp. 22721-22731).
>
> 3.  Beyer, L., Izmailov, P., Kolesnikov, A., Caron, M., Kornblith, S., Zhai, X., ... & Pavetic, F. (2023). Flexivit: One model for all patch sizes. In Proceedings of the IEEE/CVF Conference on Computer Vision and Pattern Recognition (pp. 14496-14506).
>
> 4.  Li, K., Li, X., Wang, Y., He, Y., Wang, Y., Wang, L., & Qiao, Y. (2025). Videomamba: State space model for efficient video understanding. In European Conference on Computer Vision (pp. 237-255). Springer, Cham.
>
> 5.  Wang, L., Huang, B., Zhao, Z., Tong, Z., He, Y., Wang, Y., ... & Qiao, Y. (2023). Videomae v2: Scaling video masked autoencoders with dual masking. In Proceedings of the IEEE/CVF Conference on Computer Vision and Pattern Recognition (pp. 14549-14560).

---

> ### Comment · Reviewer_gT1y · 2024-11-27
>
> I tried so hard to figure out the mathematical symbols you used even after your correction.
>
> There are still areas of confusion. For instance, if you denote $ S $ as the number of spatial tokens, what does $ L $ represent? Does it indicate the same thing? When you said you have made the change, please look at lines 193-209 and lines 255-259, you are still using $S$!
>
> Additionally, you use $ s $ as a superscript to $ e$ to suggest (s)patial embeddings. It is more conventional to use lowercase letters to denote spatial indices. Furthermore, in $ \textbf{p}_s $, $ s $ is a subscript, which does not align with its earlier usage. Why not maintain such consistency throughout?
>
> If $ L $ is defined as the number of spatial tokens, it would be more logical to use $ l = \{1, 2, 3, \ldots, L\} $ to denote the indices instead of $ i $, which is less unnecessarily confusing. Instead, you denote $ l $ as the spatial token itself, further adding to the confusion.  **Use boldface letters to denote your vectors or multi-dimensional variables !**
>
> For a paper with such inconsistent and unclear usage of mathematical symbols, these issues detract significantly from the main ideas and hinder readability.
>
> Regarding Figure 2, while it has improved slightly, there are still notable issues. For example, in the bottom-left corner, the three lines are all linked to the same patch, which is visually misleading.

---

> > ### Author Response · Authors · 2024-11-27
> >
> > We apologize for the confusion around our paper stemming from the choice of mathematical notation. Firstly, we did miss some notations that should have been updated after our first rebuttal, so we have taken extra precaution to proof every new change we have made in the updated PDF.
> >
> > To address your concern regarding notational convention, we have decided to adopt the original notation of ViT [1] due to its compatibility with our work. In our earlier rebuttal, we said we would follow VideoMamba's notation as that paper was the most relevant to our work and to any future readers. However, VideoMamba's notation is extremely simplified and sparse since they assume the reader is already familiar with the preprocessing stage for videos (e.g. patchification, adding positional embeddings, etc.). Furthermore, their notation was inconsistent with ours regarding superscripts and subscripts. Upon further consideration, we decided adopting ViT's notation was best as our method heavily revolves around the preprocessing stage for videos, and explicitly describing and assigning variables to each component of our method is imperative for improved clarity. As ViT labels almost every component needed in our work, **we have taken the reviewer's advice and improved the notation, bolded all multi-dimensional vectors, clarified the variables used for indexing, and have followed consistent notation throughout the paper**. All of these changes are highlighted in blue in the updated PDF, and for easier reference and your convenience, we provide a list of the final notations below:
> >
> > 1. Notations Taken from ViT:
> >      - Input Video: $\mathbf{x}$
> >      - Patch Size: $P$
> >      - Number of Spatial Patches (in each frame): $N$
> >      - Each Individual Patch $\mathbf{x}_n$ (indexed by $n \in \\{1, 2, \cdots, N\\}$
> >      - Embedding for each Patch: $\mathbf{E}^s_n$ (indexed by $n$ from above)
> >      - Spatial Positional Embedding: $\mathbf{E}_{pos}$
> >      - Feature for a Single Frame: $\mathbf{z}^s$
> > 2. Notations We Needed to Introduce
> >      - Temporal Positional Embedding: $\mathbf{E}_{tpos}$
> >      - Final Feature for a Frame after adding Temporal Position: $\mathbf{z}^t$
> >
> > Regarding Figure 2, we have fixed the visual issue raised by the reviewer. However, without any additional feedback, we are not sure what else needs to be added to improve the figure. We believe it now properly conveys the main idea and innerworkings of our method.
> >
> > We thank the reviewer for their comments as we believe this has lead to increased clarity in our paper, which should better highlight the main ideas, contribution to the ICLR community, and better foster reproducibility. We hope that with these implemented changes, in conjunction with the strengths mentioned by the other reviewers (such as extensive experiments, visualizations, and the advancement of the application of VideoMamba), that the reviewers feel the same way.
> >
> >
> > 1. Dosovitskiy, A. (2020). An image is worth 16x16 words: Transformers for image recognition at scale. arXiv preprint arXiv:2010.11929.

---

> > > ### Author Response · Authors · 2024-12-02
> > >
> > > Dear Reviewer gT1y,
> > >
> > > With a little over 12 hours left before the end of the reviewer rebuttal period, we would encourage you to read our response to your constructive feedback above. We believe your comments have helped improve the clarity of the paper and hope that we have finally addressed all of your concerns. We look forward to hearing back from you!
> > >
> > > Sincerely,
> > > Authors of Submission #2470

---

> > > > ### Author Response · Authors · 2024-12-03
> > > >
> > > > Dear Reviewer gT1y,
> > > >
> > > > With the reviewer discussion period quickly coming to a close, we again ask if your concerns have been sufficiently addressed. We have made the requested changes regarding the notation used in the paper and believe the issue has been resolved, especially with the other reviewers giving good scores regarding writing quality/clarity. Do let us know if you have any more comments! Thank you.

---

### Author Response · Authors · 2024-11-21
**Addressing Shared Comments Among Most Reviewers (1/2)**

There were 3 common concerns/comments brought up by 2 or more reviewers. We decided to answer them here for easier discussion and visibility. Individual questions/concerns are still answered in a direct reply to each reviewer, so do please refer to those as well. The 3 common concerns we will answer here are as follows:

1. **Can you show results on larger datasets than the ones in the paper? (R1, R2, & R3)**
	- We acknowledge that UCF-101, HMDB51, COIN, and Breakfast are not the largest action recognition datasets available (the largest in this list being COIN with ~$46K$ total clips). To provide some context, the motivation behind the choice of these datasets were for readily available baselines to compare with, and more importantly, the datasets' specific content. As noted in Lines (300-303), UCF-101 and HMDB51 are short-video action datasets, meaning a model can deduce the action being performed in only a couple frames. Contrastively, COIN and Breakfast are long-video action recognition datasets, meaning the entire action takes place over many minutes, requiring many more frames and long-range video understanding to perform well on.
	- Regardless, to further show that our results still hold on larger datasets, we perform video retrieval on Kinetics700 with vanilla VideoMamba and StretchySnake (reminder that evaluation details can be found in Section A.2). We are $\textbf{not}$ performing any training, we are simply performing video retrieval with the same models discussed in the main paper. We perform these video retrieval experiments under two different settings - only evaluating on novel classes in K700 that are not present in K400 since K400 is a subset of K700 (resulting in ~$202K$ training videos, ~$15K$ testing videos), and evaluating on all labels in K700 (resulting in ~$520K$ training videos, ~$34K$ testing videos). Even with the massive increase in data as compared to the aforementioned datasets, we show that StretchySnake's training method is scalable and still beats out vanilla VideoMamba by a significant amount, especially on unseen data.

## Kinetics400 & Kinetics700 Video Retrieval Results

| Dataset | Model  | New K700 Classes Only | All Classes |
|  :-----:  |  :----: |  :-----:  |  :---------: |
| K400 | VideoMamba | - | $78.0$
| K400| StretchySnake | - | $78.2$ (+$0.2$)
| K700 | VideoMamba | $51.5$ | $51.3$
| K700| StretchySnake | $59.4$ (+$7.9$) | $57.0$ (+$5.7$)


2. **What are the results like if you train a model flexibly from scratch, rather than loading pre-trained weights first? (R2 & R3)**
	- Firstly, it is important to note that starting with pre-trained weights does not significantly affect our results as the baseline vanilla VideoMamba we compare against is initialized with the same exact weights. Therefore, both StretchySnake and vanilla VideoMamba benefit from the pre-training, yet we still obtain massive performance gains with our proposed training method. We also discuss in our first response to Reviewer TETD that the upstream weights we load before training is standard protocol in previous papers and is somewhat necessary due to VideoMamba's sensitivity during training (as discussed in the original VideoMamba paper).


	- More importantly, we still think investigating how flexible training from scratch can affect performance is a valuable question. Thus, following Reviewer TETD, we train both a vanilla VideoMamba and a flexible VideoMamba "nearly from scratch" on the small UCF-101 dataset. It is "nearly from scratch" as we still load the ImageNet weights of VideoMamba, since training VideoMamba (and video models in general) from complete scratch on videos without first training on images has been shown to be suboptimal and difficult [1]. We provide the results in the table below, where we show that "from scratch" StretchySnake still performs marginally better than "from scratch" vanilla VideoMamba. However, we motivate throughout our paper that flexible training helps mostly with unseen data and generalization to other domains, hence our focus on video retrieval results on both short- and long-term action recognition datasets. Thus, we also perform video retrieval on the same datasets in the paper, and again we see that even training flexibly from scratch still leads to good performance gains in other downstream domains. We hope this further supports the validity and benefit of our contribution.



## Video Retrieval Results after Training from Scratch on UCF101

| Model | UCF101 | HMDB51| Breakfast | COIN |
| --- | :------: | :-----------: | :---------------: | :---------------: |
| VideoMamba | $92.0$ | $47.5$ | $22.9$ | $50.1$ |
| StretchySnake | $92.7$ (+$0.7$) | $49.4$ (+$1.9$) | $25.1$ (+$2.2$) | $52.0$ (+$1.9$) |


1. Arnab, A., Dehghani, M., Heigold, G., Sun, C., Lučić, M., & Schmid, C. (2021). Vivit: A video vision transformer. In Proceedings of the IEEE/CVF international conference on computer vision (pp. 6836-6846).

---

> ### Author Response · Authors · 2024-11-21
> **Addressing Shared Comments Among Most Reviewers (2/2)**
>
> 3. **Does flexible training have benefits to other downstream video understanding tasks? (R2 & R3)**
> 	- We would like to reiterate that our choice of evaluation datasets in the paper was not purely based on dataset size (addressed in Q1), but on the content/domain of each dataset. We show that StretchySnake improves over VideoMamba in both short- and long-video action datasets, highlighting its adaptibility to different temporal contexts. To further support this point, we also provide additional experiments here on Diving-48, a fine-grained action recognition dataset consisting of competitive diving videos. It contains ~$18K$ untrimmed videos and the real difficulty comes with distinguishing between the 48 very fine-grained classes, as the difference between two classes can be identical except for whether the diver enters the water head-first or feet-first. Thus the length of the temporal context is not the only difficulty, but being able to distinguish important keyframes as well. We perform video retrieval experiments as well as fine-tuning results on this dataset and again show that StretchySnake outperforms VideoMamba in all settings. Note that the video retrieval results are low in general due to the highly fine-grained actions in this dataset, which Kinetics-400 pre-training does not fully support. We see that after fine-tuning, both models are able to perform on this temporally sensitive dataset, with StretchySnake still outperforming VideoMamba by a significant margin. Therefore, we have exhibited StretchySnake's superiority in coarse-grained action recognition (both short- and long-video action recognition), fine-grained action recognition (Diving-48), and on large-scale action recognition datasets (K400 and K700 from Q1). We hope this provides sufficient evidence to support our claim that st-flexibility can provide significant benefits to the field of action recognition.
>
> ## Results on Diving-48
> | Model | Video Retrieval | Finetune |
> | :------: | :------: | :------: |
> | VideoMamba | $8.1$ | $81.0$
> | StretchySnake | $14.9$ (+$6.8$) | $83.3$ (+$2.3$)

---

### Meta-Review · Area_Chair_PN8G · 2024-12-20

**Metareview:**

The paper proposes StretchySnake, a spatio-temporal flexible training method for action recognition which based on the latest VideoMamba architecture. Experiments are done on UCF-101, HMDB-51, Breakfast, and COIN with good results compared with vanilla VideoMamba.

On the positive side:
* **Positive 1**: The proposed method seems simple and yet effective once pre-training on Kinetics and evaluated on small datasets such as UCF-101, HMDB-51, Breakfast, and COIN (with good improvements) thanks to the flexible training input (can be also considered as augmentations).

On the negative side:
* **Negative 1** There is no experiments done on large-scale datasets (see details below).
* **Negative 2** There is no direct comparison with latest methods based on CNNs or ViTs on the large-scale datasets.

## Further explanation on Negative points.
* **Negative 1** Although in the authors response, they provided additional experiments on Kinetics-700. However, the experimental setup is video retrieval (where the paper claims action recognition as in its title) and it requires no training while the proposed method is claimed to be flexible **training** approach.
* **Negative 2** There is one place in discussion between the authors and reviewer oh1A, the authors provided results on K-400 with both video-retrieval and classification results. Note that this experiments show very small differences (0.2 and 0.4) between vanilla VideoMamba and the proposed method, this may indicate that the flexible training may be beneficial only to small size datasets and when we have enough data, this *augmentation* (or flexible input sampling) is not very useful. Furthermore, the proposed method only compare with vanilla Video Mamba, not the latest methods of CNN-based or ViT-based.

## Final Recommendation
Although the proposed method has some potential, the current paper does not have experiments on a large-scale dataset (K-400, K-600, or K-700) with **a fair setting** (e.g., classification not a retrieval and compare with all approaches not just VideoMamba). Additionally, it would be more flexible if the proposed approach can be trained from scratch (or at least pre-trained with ImageNet only) to have fair comparison and flexibility for model / training adoption.

With the above observation, AC recommends not to accept the paper in its current form and encourages the author(s) to improve the paper and submit it to future conferences.

**Additional Comments On Reviewer Discussion:**

**The context of discussions**. The discussion between reviewers and authors and among reviewers are considered. We also consider the private message from the authors and take all into consideration. We agree that the rating of 1 from reviewer gT1y is too harsh. Thus, the main reason to reject this paper is not about the novelty (as raised by reviewer gT1y), but about the negative points mentioned above. We believe that it is minimum requirements for a paper proposing any new approach on action recognition / video classification should have experiments at least on one to two large datasets such as Kinetics (any variant of K-400, 600, 700) and Something-Something-v2 with notable improvements (0.2-0.4 improvements are not significant enough).

**Minor comments**
* The authors stated that they picked datasets such as Breakfast and COIN because these are long-form videos. However, COIN videos mean length is ~2.36 minutes which is not long.
* Furthermore, most of selected datasets used in this paper are outdated: UCF-101 (published 2012), HMDB51 (published 2011), Breakfast (published 2014), while video understanding is a fast growing and moving field with many latest benchmarks such as Kinetics, Something-Something, EPIC-Kitchen.

---

### Decision · Program_Chairs · 2025-01-22

Reject